# Res-CN: hydrometeorological time series and landscape attributes across 3254 Chinese reservoirs

Youjiang Shen[1], Karina Nielsen[2], Menaka Revel[3], Dedi Liu[4], Dai Yamazaki[1,3]

[1]Department of Civil Engineering, Graduate School of Engineering, The University of Tokyo, Tokyo, 113-0033, Japan
[2]DTU Space, National Space Institute, Technical University of Denmark, 2800, Kongens Lyngby, Denmark
[3]Global Hydrological Prediction Center, Institute of Industrial Science, The University of Tokyo, Tokyo, 153-8505, Japan
[4]State Key Laboratory of Water Resources Engineering and Management, Wuhan University, Wuhan, 430072, China

*Correspondence to*: Youjiang Shen (yjshen2022@rainbow.iis.u-tokyo.ac.jp)

**Abstract.** Dams and reservoirs are human-made infrastructures that have attracted increasing attentions because of their societal and environmental significance. Towards better management and conservation of reservoirs, a dataset of reservoir-catchment characteristics is needed, considering that the amount water and material flowing into and out of reservoirs depends on their locations on the river network and the properties of upstream catchment. To date, no dataset exists for reservoir-catchment characteristics. The aim of this study is to develop the first database featuring reservoir-catchment characteristics for 3254 reservoirs with storage capacity totaling 682,595 km$^3$ (73.2% reservoir water storage capacity in China), to support the management and conservation of reservoirs in the context of catchment level. To ensure a more representative and accurate mapping of local variables of large reservoirs, reservoir catchments are delineated into full catchments (their full upstream contributing areas) and intermediate catchments (subtracting the area contributed by upstream reservoirs from full upstream of the current reservoir). Using both full catchments and intermediate catchments, characteristics of reservoir catchments were extracted, with a total of 512 attributes in six categories (i.e., reservoir and catchment body characteristics, topography, climate, soil and geology, land cover and use, and anthropogenic activity characteristics). Besides these static attributes, time series of 15 meteorological variables of catchments were extracted to support hydrological simulations for a better understanding of drivers of reservoir environment change. Moreover, we provide a comprehensive and extensive reservoir data set on water level (data available for 20% of 3,254 reservoirs), water surface area (99%), storage anomaly (92%), and evaporation (98%) from multisource satellites such as radar and laser altimeters and images from Landsat and Sentinel satellites. These products significantly enhance spatial and temporal coverage in comparison to existing similar products (e.g., 67% increase in spatial resolution of water level and 225% increase in storage anomaly) and contribute to our understanding of reservoir properties and functions within the Earth system by incorporated national or global hydrological modeling. In situ data of 138 reservoirs are employed in this study as a valuable reference for evaluation, thus enhancing our confidence in the data quality and enhancing our understanding of accuracy of current satellite datasets. Along with its extensive attributes, the Reservoir dataset in China (Res-CN) can support a broad range of

applications such as water resources, hydrologic/hydrodynamic modeling, and energy planning. Res-CN is on Zenodo through https://doi.org/10.5281/zenodo.7664489 (Shen et al., 2022a).

## 1 Introduction

The role of reservoirs in the hydrological and biogeochemical cycles is closely tied to their characteristics of water surface area, water level, evaporation, and storage variation. In addition, the amount and rate of water and materials flowing into and out of reservoirs depends on their location in the river network, reservoir upstream catchment attributes (e.g., catchment size, topography, geology, soil, and land cover) as well as meteorological variables (e.g., precipitation, and temperature). An explicit spatial knowledge of all these characteristics (see in Fig. A1) is crucial for

determining surface water availability and modulating water flux interactions among various Earth system components, including terrestrial water storage dynamics (Busker et al., 2019; Chaudhari et al., 2018); terrestrial carbon cycle (Marx et al., 2017); geochemical cycle (Maavara et al., 2020); surface energy budget (Buccola et al., 2016); climate-related effects (Boulange et al., 2021); and alterations in the hydrological and ecological processes such as sediment reduction (Li et al., 2020), degradation of water quality (Barbarossa et al., 2020), land use changing pattern

(Carpenter et al., 2011), and fish biodiversity decline (Ngor et al., 2018). Therefore, to fully uncover the functioning of reservoirs for better scientific studies and water resources managements, it is essential to develop a comprehensive publicly available reservoir data set in the context of growing interest of reservoir studies and water managements.

China is the world's most populous country that has undergone an impressive average GDP growth rate of 10% over the past two decades (Gleick, 2009). Meanwhile, it has simultaneously experienced notable expansion of irrigation and

encountered challenges arising from limited water resources, frequent floods, and droughts (Wang et al., 2020). To ensure water security, reservoir construction is proliferating across the country. As of 2015, China had constructed approximately 98,000 reservoirs and dams, including almost 40% of the world's largest dams (Song et al., 2022). The world's largest clean energy corridor, comprised of six mega hydropower dams, is newly formed in China. Despite these developments, there remains a data gap regarding the surface water dynamics and upstream attributes of these

reservoirs at the catchment level.

In recent years, multiple efforts have been made to produce reservoir inventories, including those of China. For the inventories of water surface area, water level, evaporation, and storage anomaly, there are different research projects and studies producing satellite datasets for reservoirs at regional and global scales (Crétaux et al., 2011; Birkett et al., 2011; Schwatke et al., 2015; Markert et al., 2019; Tourian et al., 2022; Tortini et al., 2020; Zhao & Gao, 2018; Liu et al.,

2021; Donchyts et al., 2022; Vu et al., 2022; Tian et al., 2022). However, information of reservoir characteristics is still insufficient and scarce across different regions. The majority of them are devoted to developing a particular type of reservoir data set for selected globally distributed reservoirs (Gao et al., 2012; Zhao et al., 2022). For example, Zhao et

al. (2019) constructed the long-term monthly evaporation time series for 721 reservoirs in the U.S. by using four meteorological forcings and Landsat-based images. Other remotely sensed datasets such as storage anomalies from (e.g., Busker et al., 2019; Hou et al., 2022) and water surface areas from (Klein et al., 2021) are not publicly accessible. Remotely sensed reservoir datasets, estimated by different researchers, are usually not consistent on the aspects of target water bodies and data sources, which make it difficult to provide consistent baseline of reservoir characteristics for a specific region or country. For example, Khandelwal et al. (2022) generated monthly water surface areas over the new lake polygons while other studies produced water surface area time series over the reservoir and/or lake shapefiles from some existing databases such as GRanD (Lehner at al., 2011) and HydroLAKES (Messager at al., 2016). In addition, there is no systematic assessment of whether reservoir water levels or water surface areas from previous studies and databases agree with one another, as shown in this study by many reservoirs whose in situ measurements are available. Here, we list remotely sensed databases containing Chinese reservoirs in Supplementary Table S1. Only a small number of reservoirs are available from these databases. In three popular altimetry-based reservoir datasets (Hydroweb, G-REALM, and DAHITI), there are approximately 30 Chinese reservoirs. Although Shen et al. (2022b) used GRanD reservoir shapefiles and multisource altimeters to generate a data set of water level, water surface area, and storage anomaly for 338 Chinese reservoirs during 2010-2021, there is still room for additional complements to the existing databases in its spatial and temporal coverage.

In addition to the time series of reservoir datasets described above, reservoir upstream catchment attributes (e.g., climate, geology & soil, topography, land cover, and anthropogenic activity characteristics) are also important as reservoirs collect materials from upstream catchments. These attributes affect the water balance and water quality of a reservoir, such as temperature, dissolved oxygen, and turbidity (Yang et al., 2022). Moreover, the limnological properties of one reservoir have the potential to impact other reservoirs through the transfer of water mass, nutrients, energy, and sediments via connecting rivers, as previously demonstrated in studies by Huziy and Sushama (2017) and Stieglitz et al. (2003). Thus, researchers can better understand catchment-level landscape limnology by incorporating these attributes (Soranno et al., 2010). The values of these catchment-level attributes are also proved in the Catchment Attributes and MEteorology for Large-sample Studies (CAMELS) introduced by Addor et al. (2017) and follow-up studies such as CAMLES-CL, CMALES-BR, CAMLES-GB, (Alvarez-Garreton et al., 2018; Chagas et al., 2020; Coxon et al., 2020), LamaH-CE (Klingler et al., 2021), CCAM (Hao et al., 2021), LakeALTAS (Lehner et al., 2022), as well as the works by Chen et al. (2022) and Liu et al. (2022). However, there is a data gap of reservoir-catchment characteristics in China, and even the geometric boundaries of reservoir upstream catchment, which hindered the spatially explicit applications of such catchment information. Furthermore, allocating reservoirs on river network is also valuable for river models incorporating reservoirs as reservoir datasets and river network datasets are usually developed independently, and they are not well corresponding and could cause some issues when integrating reservoirs in river model.

In light of above, we build upon these existing studies and datasets to produce a new publicly available comprehensive and extensive reservoir dataset, Res-CN (Reservoir-ChiNa). It is based on the latest global reservoir shapefiles from GeoDAR v1.1. Additionally, we allocate reservoirs on the MERIT Hydro (Yamazaki et al., 2019) to delineate reservoir catchments into two categories: full catchments and intermediate catchments. 512 catchment-level attributes for 3254 reservoirs are generated at full catchments and intermediate catchments from a wide range of satellite-, reanalysis-,

and in-situ based data. Besides, time series of reservoir states (i.e., water level, water surface area, storage anomaly, and evaporation) are extracted from multiple altimeters, Landsat and Sentinel images and other satellites, acting as a key supplement to existing products owing to its significantly enhanced spatial and temporal coverages. In situ data of 138 reservoirs are employed in this study as a valuable reference for evaluation, thus enhancing the confidence in the data quality and enhancing the understanding of accuracy of current satellite datasets. Our codes, in Python/R/GEE

(Google Earth Engine), are freely available and open source. The code can be applied to individual applications, other areas, and can further enrich the inventory if new data available. Results of this study facilitated managements of reservoirs and relevant studies such as hydrological modeling, environmental studies, and climate research in the spatially explicit context of reservoir catchment-level (Galelli et al., 2022; Dang et al., 2020).

## 2 Data sources and methods

China has more than 98,000 reservoirs and dams across different topographic regions and landscapes (MWR, 2016). However, most of them are unmapped (polygons and georeferenced coordinates not available) and only described with standard attributes. Thus, in this study, we focused on reservoirs which are mapped and available from the newest global GeoDAR database (Wang et al., 2022). GeoDAR v1.1 provides global reservoir shapefiles and their attributes such as storage capacity, reservoir purpose and installed capacity. Reservoirs are mostly clustered in Yangtze and Pearl

River basins (Fig. 1) and vary greatly in size, capacity, and purpose.

As we aim to create a comprehensive reservoir data set in China, our workflow required multiple steps and geospatial techniques. Here, we detail the data and methodologies that is applied to create water level, water surface area, storage anomaly, evaporation, upstream catchment boundaries and catchment-level characteristics. The flowcharts and source datasets are provided in the Figures A2-3, Figures S1-3, and Table S2-8.

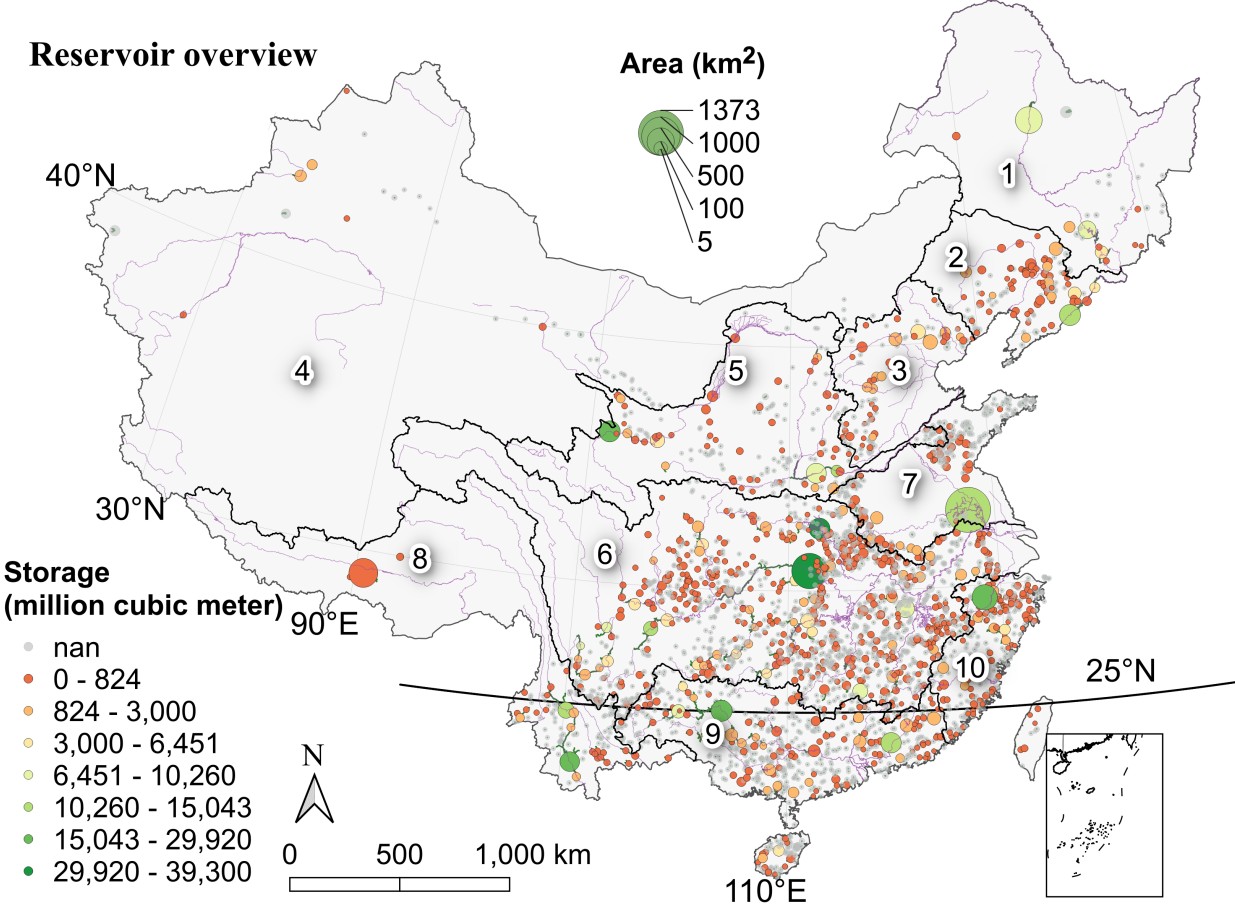

**Figure 1.** Overview of the reservoirs contained in Res-CN, and the dams with storage capacity (circle color, nan means not available) and water surface area of reservoirs (circle size). The black lines indicate the boundaries of the ten river regions within Res-CN. Numbers (1-10) indicate: 1—Songhua River; 2—Liao River; 3—Hai River; 4—Northwest River; 5—Yellow River; 6—Yangtze River; 7—Huai River; 8—Southwest River; 9—Pearl River; and 10—Southeast River regions.

## 2.1 Data and methodology for generating reservoir water level

Water level time series for the Res-CN reservoir are derived from various satellite altimeters: Sentinel-3A, Sentinel-3B, Jason-3, ICESat-2, CroySat-2, and SARAL/AltiKa. Each altimeter has different repeat cycles, geographical coverages, retracking algorithms, and measurement accuracies (Table S2). Apart from the official algorithms in their source products, we implemented PPCOG (primary peak center of gravity) and NPPTr[0.5/0.8] (narrow primary peak with a 0.5 or 0.8 threshold value) algorithms into Sentinel-3, CroySat-2, SARAL/AltiKa to derive range measurements (Shen et al., 2022b). Range measurements are corrected using the atmospheric and geophysical corrections from their source products (Table S2), and then used to determine water level of each sample. We reference the height to the EGM2008 geoid (Pavlis et al., 2012).

For the construction of reservoir water levels, we carried out the following steps to process single satellite altimetric heights from each retracking algorithm (Fig. S1):

- Extraction of the altimetric data within the GeoDAR reservoir shapefile.
- The Global Surface Water Explorer was used to select altimetric data for which water occurrence is greater than 10% (Zhang et al., 2020).
- Outlier removal for each pass using the MAD method (median of absolute deviation).
- We remove outliers from altimeter data with heights more than 20 meters from the DEM (for reservoirs with large variations, we set a threshold of 40 meters).
- Construction of time series using the R package "tsHydro" (Nielsen et al., 2015).

Through these steps, each satellite altimeter's SR (standard rate) water level time series with different retracking algorithms were produced. A single satellite altimeter's repeat period and spatial sampling results in a low resolution for SR products. For example, with Sentinel-3A ground tracks spaced 104 km apart at the equator, it may be possible to obtain altimetric data on 684 GeoDAR reservoirs in China. For increased resolution and to overcome the limitations of single satellite altimeter spatial and temporal sampling, we generated HR (high rate) water level time series products by integrating single-satellite SR products. Note that we used altimetric observations from multi-mission using the retracking algorithm with the smallest RMSE (root-mean-square error) value calculated with in situ water level. To eliminate systematic differences between satellites, we used two methods: the first method is by directly eliminating the mean water level differences between satellites and is applicable to satellites with sufficient overlap periods; the second method is to estimate satellite bias using reservoir water areas. The bias was estimated by minimizing the two-dimensional cost function of area-water level coordinates using the Gauss-Helmert method (Fig. S1). To evaluate the altimetric data quality, we calculated the standard deviation (SD) of altimetric observations and RMSE values against in situ water level and three other similar existing products from Hydroweb, DAHITI, and G-REALM wherever available. Data point precision is determined by SD, whereas accuracy is determined by RMSE. RMSE is calculated by comparing water level anomalies between gages and satellites. In situ water level of 99 reservoirs from 2015 to 2021 are used to validate our dataset.

## 2.2 Data and methodology for generating reservoir water surface area

Reservoir water surface areas can be extracted from an available global inland water dataset like the SWBD (SRTM Water Body Data, NASA JPL, 2013), the GIEMS (Global Inundation Extent from Multiple Satellites, Papa et al., 2010b), GSW, DAHITI, Hydroweb, Hydrosat, Bluedot Observatory and studies from Tortini et al. (2020) and Shen et al. (2022b). The derived reservoir water surface area estimations are limited by the spatial coverage and accuracy restrictions of the initial dataset. As such, three available global water surface area products are developed by using algorithms that reclassify contaminated pixels as water, i.e., the GRSAD (Zhao & Gao, 2018), the RealSAT (Khandelwal et al., 2022), and

areas of medium-small reservoirs by Donchyts et al. (2022). These products cover only a portion of the reservoirs we studied (e.g., 908 overlapping reservoirs between GRSAD and our product) and use different algorithms and source datasets (e.g., RealSAT and GRSAD use only Landsat). As a part of this study, we employed the algorithm developed by Donchyts et al. (2022) to generate reservoir water surface areas by using Landsat and Sentinel-2 images. The algorithm has been applied to map water surface areas in 768 reservoirs of different sizes and climate zones located in Spain, India, South Africa, and the USA, and there is strong evidence to suggest that it performs well in this regard (Donchyts et al., 2022). For a given reservoir, the procedures are as follows:

- A selection of cloudy satellite images intersecting the reservoir shapefile is made.
- Based on the global cloud frequency dataset (Wilson, 2016), filter out satellite images that are completely covered by clouds and correct the remaining images as follows.
- Calculating the NDWI (normalized difference water index).
- A Canny edge detection algorithm is used for detecting water/land edges and defining sampling areas around them (Donchyts et al., 2016).
- Utilizing the Otsu algorithm (Markert et al., 2020) to determine the optimal threshold value, then obtaining the water mask based on samples of NDWI values collected within the sampling area.
- Sampling the water occurrence along the edges to eliminate falsely detected water (water pixels that were not water).
- Water occurrence is clipped at a certain occurrence value and combined with water mask to obtain the final water mask.
- Using a quantization-based temporal outlier filter to remove any errors from reservoir waters.

Using these procedures, we generated monthly reservoir water surface areas during 1984-2021. We evaluated the data quality by comparing it to in-situ water levels, altimetry (HR and SR products) whenever available, and previously available products from GRSAD and RealSAT. The indicators of data quality were rBIAS (relative bias), CC, and rRMSE (relative RMSE).

**2.3 Data and methodology for generating reservoir storage anomaly**

Res-CN estimates of reservoir storage anomaly are based on (1) satellite-based water levels and water surface areas, and (2) water surface areas and DEMs (digital elevation models) (Fig. S2). The basis of these two approaches is a reconstruction of the hypsometry curve (water surface area-level model) using overlapping records of water level and water surface area or DEM. Assuming five models (linear, polynomial, exponential, power, and logarithmic) can be used to describe hypsometry curves, we selected the model with the highest $R^2$ value as the reservoir's hypsometry curve. We followed the following steps for reservoirs with both water levels and water surface areas records:

- Using the average of all altimetric measurements in a month to calculate the monthly reservoir water level.

- A scatterplot of monthly water level and water surface area is constructed, and errors are eliminated from the scatterplot.

- A parametric approach is used to generate the hypsometry curve (i.e., the water surface area-level model).
- Estimating the gap measurements of water levels and water surface areas by applying the reconstructed hypsometry curve.
- Calculating storage anomaly $\Delta V$ from two successive pairs of water level-water surface area measurements $(H, A)$, Eq. (1)

$$\Delta V_t = \frac{1}{2}(H_t - H_{t-1}) \times (A_t + A_{t-1}), \tag{1}$$

For reservoirs with water surface areas only, we used the DEM-based approach. The main procedures are described below:

- Generating the water surface area-level-storage model through DEM-based approach (Vu et al., 2022).
- Calculating storage anomaly by combining satellite-based water surface areas and area-level-storage model.

As a result of these steps, we determined the hypsometry curve and time series of storage anomalies for each reservoir. To evaluate the storage anomaly data, we calculate the error statistics of RMSE, NRMSE (normalized root-mean-square error), and CC (pearson correlation coefficient) values for reservoirs with in situ observations. In situ water storage of 138 reservoirs from 2015 to 2021 are used to validate our dataset.

## 2.4 Data and methodology for generating reservoir evaporation

Reservoir evaporation estimates can be extracted from two available global reservoir evaporation products produced by the Zhao et al. (2022) and Tian et al. (2022). These studies covered a portion of the reservoirs we studied and used the same robust algorithm by Zhao et al. (2019) to calculate monthly reservoir evaporation volume ($V_E$, [m$^3$/month]) using water surface area ($A$, [km$^2$]), days of a month ($days$, [-]), and evaporation rate ($E_{reservoir}$, [mm/day]) (Eq. 2).

$$V_E = E_{reservoir} \times A \times days/1000 \tag{2}$$

In this study, we applied the algorithm as well considering its satisfactory simulation performance for the evaporation rate and energy balance terms (Eqs. 3-4). It quantifies heat storage changes ($\delta U$, [MJ m$^{-2}$ d$^{-1}$]) in the Penman equation

$$E_{reservoir} = \frac{\Delta(R_n - \delta U) + \gamma \lambda (2.33 + 1.65 u_2) L_f^{-0.1}(e_s - e_a)}{\lambda(\Delta + \gamma)} \tag{3}$$

$$\delta U = \rho_w c_w h \frac{T_w - T_{w0}}{\Delta t} \tag{4}$$

where $E_{reservoir}$ is the reservoir evaporation rate (mm/d); $\Delta$ is the slope of the saturation vapor pressure curve
(kPa/°C); $R_n$ is the net radiation (MJ m$^{-2}$ d$^{-1}$); $\gamma$ is the psychrometric constant (kPa/°C); $\lambda$ is the latent heat of vaporization (MJ/kg); $u_2$ is the wind speed at 2 m height (m/s); and $L_f$ is the monthly reservoir fetch length (m); $e_s$ and

$e_a$ are the saturated vapor pressure at air temperature and the air vapor pressure (kPa), respectively; $\rho_w$ is the density of water (kg m$^{-3}$); $c_w$ is the specific heat of water (MJ kg$^{-1}$ °C$^{-1}$); $h$ is the average water depth (m); $T_w$ and $T_{w0}$ are the water column temperature at the current time step, and at the previous time step (°C), respectively; and Δt is the time step (set as 1 month in this study).

It is worth noting that this algorithm does not need parameter calibration and only requires the $L_f$, $h$, and four meteorological variables (air temperature [°C], wind speed [m/s], vapor pressure deficit [kPa], and surface downward shortwave radiation [MJ m$^{-2}$ d$^{-1}$]), for the evaporation rate calculation. In previous studies (Zhao et al., 2022; Tian et al., 2022), the TerraClimate dataset (Abatzoglou et al., 2018) has been shown to be the most appropriate meteorological dataset for reliable estimates of reservoir evaporation rates compared to other global datasets. Thus, we adopted the TerraClimate to generate meteorological data time series by averaging gridded forcing data within reservoir shapefiles. Monthly reservoir fetch values $L_f$ were calculated using (1) reservoir shapefile, (2) wind direction data from the NCEP (National Centers for Environmental Prediction), and (3) water surface area time series. The average reservoir water depths $h$ are taken from the GRanD and HydroLAKES datasets. For reservoirs not recorded in these two datasets, we determined their water depths from their water surface areas through the area-depth curves fitted to all available data. A detailed algorithm flowchart, all equations, and examples are provided in the Figure S3 and Zhao et al. (2019).

## 2.5 Data and methodology for generating reservoir catchment boundaries

In this study, two types of reservoir upstream catchment boundaries (hereafter, referred to as catchment) are defined: full catchments and intermediate catchments (Fig. 2). The full catchment is defined as a reservoir's full upstream contributing area, whereas the intermediate catchment is determined by subtracting the area contributed by upstream reservoirs from the catchment area of the current reservoir. Obviously, full catchments are independent with each other, but for reservoirs with larger catchments, they can lead to a significant loss of information such as the local features and variability. Thus, intermediate catchments, which are part of the large river networks, can complement this information and ensures more representative and accurate mapping of local variables.

To identify catchment boundaries (Fig. A2), we used an automatic outlet relocation algorithm (Xie et al., 2022) to automatically delineate a large amount of catchments. This algorithm can correct the river networks by analyzing the gradients of flow accumulations along the rivers and can rapidly delineate catchments (Xie et al., 2022). More importantly, it has been intensively validated in 1,398 catchments of varying size and geographic regions, showing 94.1% of catchments were correctly delineated. This algorithm requires the flow directions and gauge locations as input. In this study, we used flow directions from the MERIT Hydro and dam locations from the GeoDAR dataset. MERIT Hydro is a new global hydrography map that has a fine resolution of 90 m and shows good performance in terms of river basin shape and flow accumulation area. This algorithm can generate the full catchments of each reservoir, and we employed some additional operations to remove topology errors and obtain intermediate catchments. Firstly, we cleared the

holes to remove topology errors across full catchments. Secondly, we checked the full catchments, and removed the
unrealistic or incorrectly catchments. Thirdly, we generated intermediate catchments by removing the overlapping
areas of upstream reservoirs from the full catchment of the current reservoir using QGIS 3.24. Lastly, we fixed the
invalid geometry of intermediate catchments by eliminating geometry errors (Text S1).

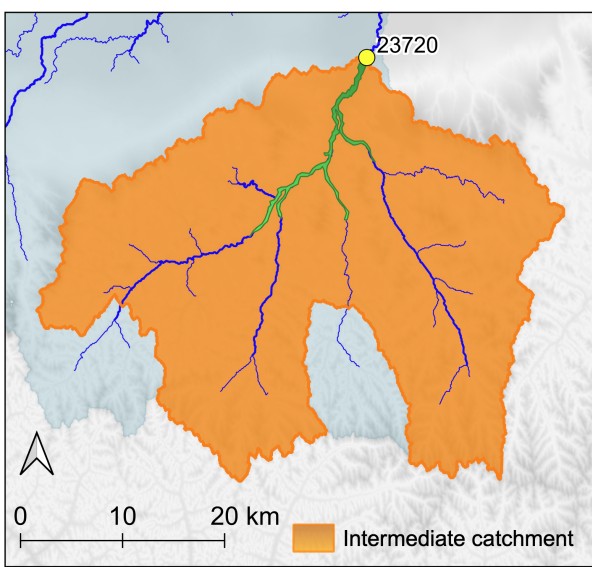

**a) Basin delineation  A | Full catchments**

**b) Basin delineation. B | Intermediate catchments**

**Figure 2.** An example of the types of catchment delineations in Res-CN. (a) Catchment delineation A: full catchments, which are
defined as the entire area contributing to a reservoir. In plot (a), full catchment of reservoir 23720 overlaps with that of reservoir
3205 and that of 6651. (b) Catchment delineation B: intermediate catchment. In plot (b), all upstream contributing areas of the
upstream reservoirs (3205 and 6651) are removed from the full catchment of reservoir 23720, thus, we get the intermediate
catchment of reservoir 23720 (in black boundary). Background in light blue indicates other catchments not shown in this example.
Source of background: MERIT Hydro and MERIT DEM (Yamazaki et al., 2019).

**2.6 Data and methodology for generating catchment-level characteristics**

Catchment attributes can be categorized into six types: reservoir and catchment body characteristics, climate, geology
& soil, topography, land cover, and anthropogenic activity characteristics. The sources for different datasets were
chosen to ensure use of high-quality, most reputable, global coverage, theoretical impacts on reservoirs, and high
spatial resolution as far as possible. For example, the NSCD version 3 (National station-based climatic data set) is the
most widely used in situ meteorological dataset in China, while the MERIT DEM is a new baseline of global hydrography
map. Here, we give a brief description of data processing, while the detailed interpretations, uncertainties, and
limitations of source datasets are available in Section 3 and Supplementary (Fig. A3).

Some necessary data format conversions (e.g., netCDF to raster format) and reprojections are firstly conducted for related attribute datasets. Then, we used different methods to calculate catchment-level characteristics from raster and vector data. Statistics values were calculated for raster grids within each catchment. In the case of continuous variables such as temperature and elevation, we calculated their mean, maximum, minimum, and range values. For categorical variables such as geological maps and land use features, we calculated the percentages of each variable and determined their dominant type. The implementation for processing raster data is done in the local Python and R environments (package: rasterstats) and the GEE platform. For vectorial data such as the rivers and catchments in ESRI shapefile format, the shape features of each catchment are calculated using the local Python scripts, and other catchment attributes like stream density are determined by the ratio of the intersection area between the vectorial data and the catchment extent layer to the total catchment area. After repeating these procedures for each catchment, catchment-level attributes are prepared for both full catchments and intermediate catchments.

## 3 Results and discussion

### 3.1 Description of the Res-CN database

We here provide a summarized information on the components of the Res-CN in Table 1. Detailed descriptions of each component are shown in the following sections and Supplementary materials. In this study, we constructed reservoir-catchment characteristics for 3254 reservoirs recorded in the GeoDAR database (Wang et al., 2022) with water surface areas ranging from 0.004 and 1373.77 $km^2$ (Fig. S4), and storage capacity totaling 682,595 $km^3$ (73.2% reservoir water storage capacity in China). Using both full catchments and intermediate catchments, characteristics of reservoir catchments were extracted, with a total of 512 attributes in six categories (Table 1). Besides, time series of reservoir states such as water level and water surface area are also provided, with their comprehensive evaluation reports (i.e., statistics and figures in PDF and Excel files) based on in situ data when available. For more details, please refer to the data repository and the following sections.

**Table 1. Summary of the data provided in the Res-CN.**

| | Variable | Number of (reservoirs/catchments) | Description |
|---|---|---|---|
| **Time series of reservoir states** | Water level (SR, a total of 650 reservoirs) | 54 | From Jason-3, 10-days, 2016-2022, with 3 retracking algorithms |
| | | 192 | From Sentinel-3A, 27-days, 2016-2022, with 5 retracking algorithms |
| | | 194 | From Sentinel-3B, 27-days, 2018-2022, with 5 retracking algorithms |
| | | 215 | From ICESat-2, 91-days, 2019-2022, with 1 retracking algorithm |
| | | 347 | From CryoSat-2, 369-days, 2010-2022, with 3 retracking algorithms |

| | | 229 | From SARAL/AltiKa, sub-cycles of 15-17 days, 2016-2022, with 5 retracking algorithms |
|---|---|---|---|
| | Water level (HR) | 250 | High rate (HR) product by merging standard rate (SR) products, from 2010-2022, sub-monthly or monthly |
| | Water surface area | 3214 | Monthly from 1984-2021 |
| | Storage anomaly | 2999 | Monthly storage anomaly from 1984-2021 |
| | Evaporation | 3185 | Monthly evaporation rate and volume from 1984-2021 |
| **Catchment-level attributes** | Catchment body characteristics | 3254 full catchments, 435 intermediate catchments | Two types of reservoir upstream catchments, reservoir and catchment body attributes (Tables S9-10) |
| | Topography | Same as above | 19 attributes (Table S10) |
| | Climate data | Same as above | 11 climatic attributes and daily time series of metrological data with 15 variables from 1980-2022 (Tables S11-12) |
| | Land cover | Same as above | 23 attributes (Table S13) |
| | Soil & Geology | Same as above | 173 attributes (Tables S14-15) |
| | Anthropogenic activity characteristics | Same as above | 288 attributes (Table S16) |

### 3.2 Res-CN products for the delineated catchment characteristics

Res-CN provides 3254 full catchments and 435 intermediate catchments (Fig. 3). The median catchment size of full catchments is 294 km$^2$, with a range of 0.94 to 981,473 km$^2$. The plausibility of full catchment delineation was assessed by comparing the area of the delineated catchments with the areas of two declared references: GRanD (Lehner et al., 2011) and LakeATLAS (Lehner et al., 2022). LakeATLAS delineated upstream drainage area of more than 1.4 million lakes and reservoirs globally based on the lake pour points and the 15 arc-seconds drainage direction grids of HydroSHEDS. A similar approach was applied in the GRanD to estimate the areas of upstream catchments over 7,320 reservoirs globally. To compare Res-CN with GRanD and LakeATLAS, we spatially joined reservoir shapefiles from both datasets, matching reservoirs that overlapped for greater than 90% of their extent. Based on this subset of reservoirs, we found that catchment areas delineated in this study corresponded relatively well to catchment areas in both GRanD (CC = 0.99, n = 910) and LakeATLAS (CC = 0.91, n = 2147), which proves the reliability of our delineated catchments. Large discrepancies occur in 55 catchments, whose absolute relative error is greater than 100% (Fig. 3e, f). Small reservoirs located near confluences between rivers of different sizes are more likely to be affected by this issue, as a minor spatial mismatch can assign a reservoir to the small catchment of the tributary stream rather than the large catchment of the mainstream, and vice versa (Fig. S5). The differences in catchment delineation between these datasets result from differences in both DEM and methods for flow direction correction and depression filling and pour points correction. In this study, the widely verified MERIT Hydro flow directions at 3 arc-seconds are used, and we suggest

that cautions should be taken when using catchments with large error discrepancies with LakeATLAS, which is based on the 15 arc-seconds drainage direction grids of HydroSHED (Fig. S5a). Intermediate catchments provide information regarding the variability of local features and upstream–downstream relationship. The median catchment size of intermediate catchment is 936 km², with a range of 1 to 279,424 km².

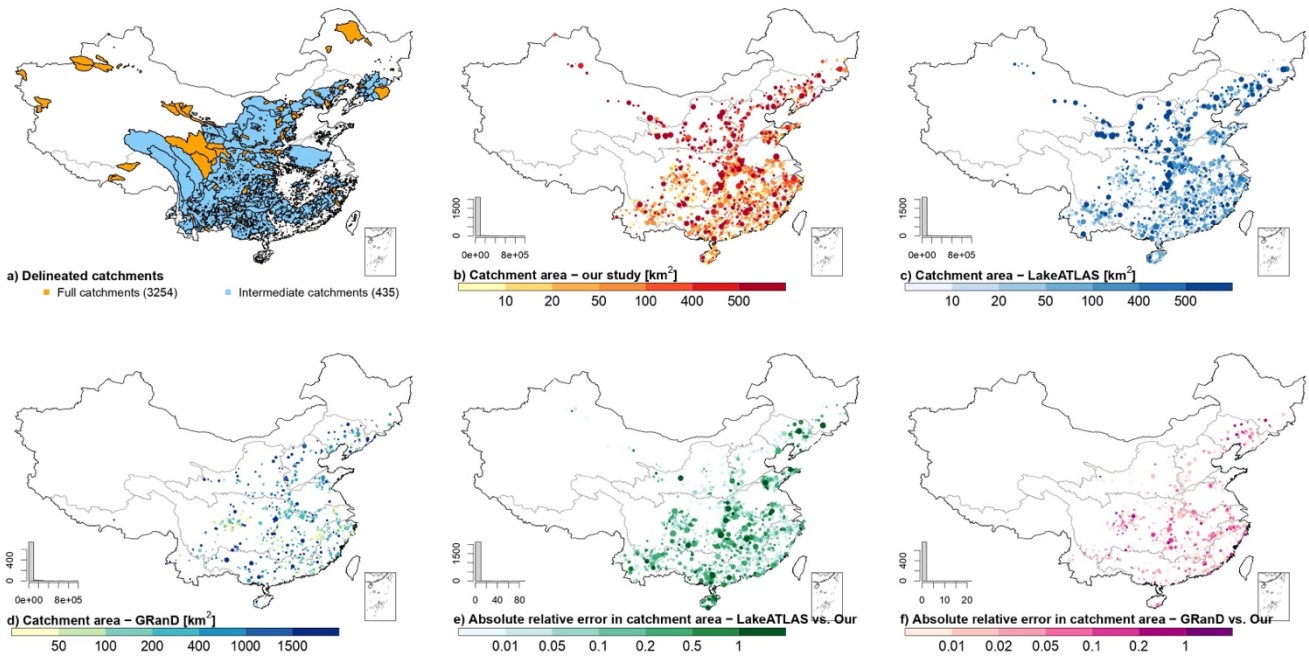

**Figure 3.** Distribution of the delineated catchments (intermediate catchments and full catchments). Each category's histogram indicates the number of basins (out of 3254). In a histogram, the X-axis represents the number of basins, while the Y-axis represents each subplot's title. Circle sizes are proportional to catchment areas.

## 3.3 Res-CN products for time series of reservoir states

### 3.3.1 Reservoir water level

There are two modes of reservoir water level time series available from Res-CN: SR and HR. Fig. 4 demonstrates their spatial coverages, data source and availability, and evaluation reports against in situ observations. Among them, over 200 reservoirs are visited by CroySat-2, SARAL/AltiKa and ICESat-2, and only 192 and 54 reservoirs are covered by Sentinel 3A and Jason 3, respectively (Table 1). Data quality was generally good for smaller RMSE values (< 0.3 m), moderate for those between 0.3 and 1.0 m, and relatively poor for those greater than 1.0 m. For each altimeter and HR product, percentage of validated reservoirs with good, moderate and poor data quality is 44%,33%, 23% (Sentinel-3A: validated in 34 reservoirs), 55%, 18%, 27% (Sentinel-3B: 22), 38%, 37%, 25% (SARAL/AltiKa: 8), 71%, 10%, 19% (ICESat-2: 31), 50%, 36%, 14% (Jason-3: 14), 22%, 56%, 22% (Cryosat-2: 27), and 25%, 73%, 2% (HR products: 84). We found that in most cases there is no notable difference in terms of RMSE values between different retracking

algorithms (Fig. S6). It should be noted that multisource altimetric measurements are merged for a specific reservoir by using the SR time series with the lowest RMSE from the retracking algorithm. Fig. 5 shows examples of HR products over a sample of six reservoirs with different areas. Single-altimetric time series capture reservoir water level dynamics

well, leading to improved temporal resolution of HR product. A cross validation of the time series against other existing databases as well as a comparison of their spatial coverage in China (Table S1 and Fig. S7) further demonstrated the advantages of our products. Water levels provided by Res-CN generally agree with those provided by exsiting similar products (Hydroweb, G-REALM, and DAHITI) with CC values exceeding 0.9, although there are some discrepancies. As an example, Res-CN time series are much denser and less noisy than Hydroweb's in most reservoirs. At the Sanhezha

reservoir, G-REALM failed to capture the clear fluctuation pattern, while large discrepancies were apparent at the Fengman reservoir in 2020 between Res-CN and Hydroweb (Fig. S7).

We should consider some limitations for further improvements despite our products' good performance and expanded spatial coverage. We provided the uncertainty information for each value of the time series in the data product file. The SD (standard deviation) estimates can quantify the accuracy of the water level along the track at the level of individual

data points (Fig. S8). Water level time series for each reservoir are available in Rec-CN as EXCELs, PDFs and detailed evaluation reports based on in situ data when available (see Section of data availability and Fig. S9). In our study, more than 80% of reservoirs with inadequate altimetric measurements are removed due to the inherent limitations of satellite altimeters. There are still challenges to delivering useful measurements for certain reservoirs along the Yangtze and Yellow rivers, and data quality is poor in terms of RMSE values regardless of reservoir size. It may be

possible to apply advanced algorithms such as machine-learning in future studies to achieve better performance regardless of whether reservoirs represent different behaviors.

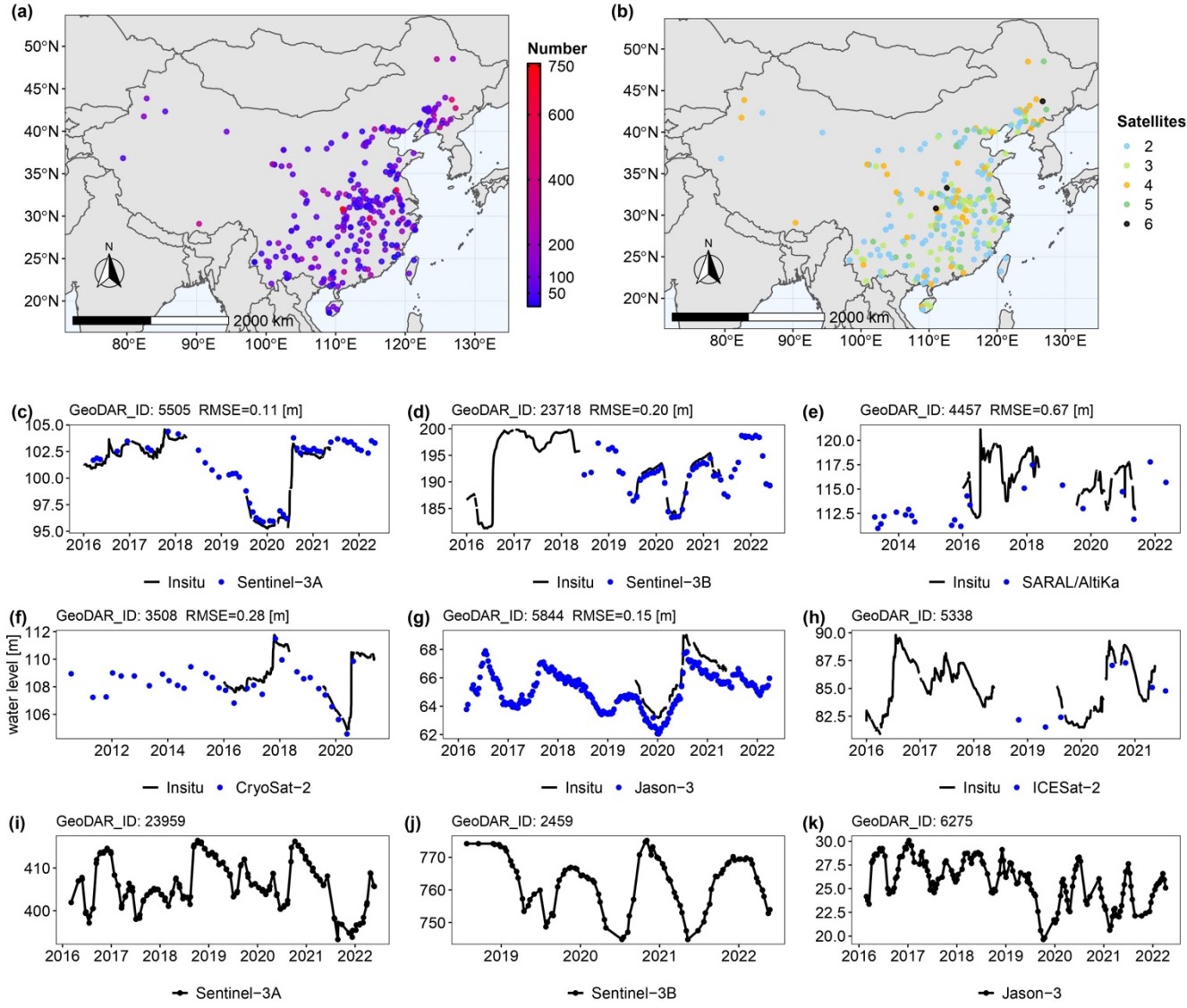

**Figure 4.** Data data availability of the altimetric reservoir water level time series (a) and number of satellites for High-rate products (b) as well as some examples to illustrate time series of Standard-rate products over nine selected reservoirs (c-k).

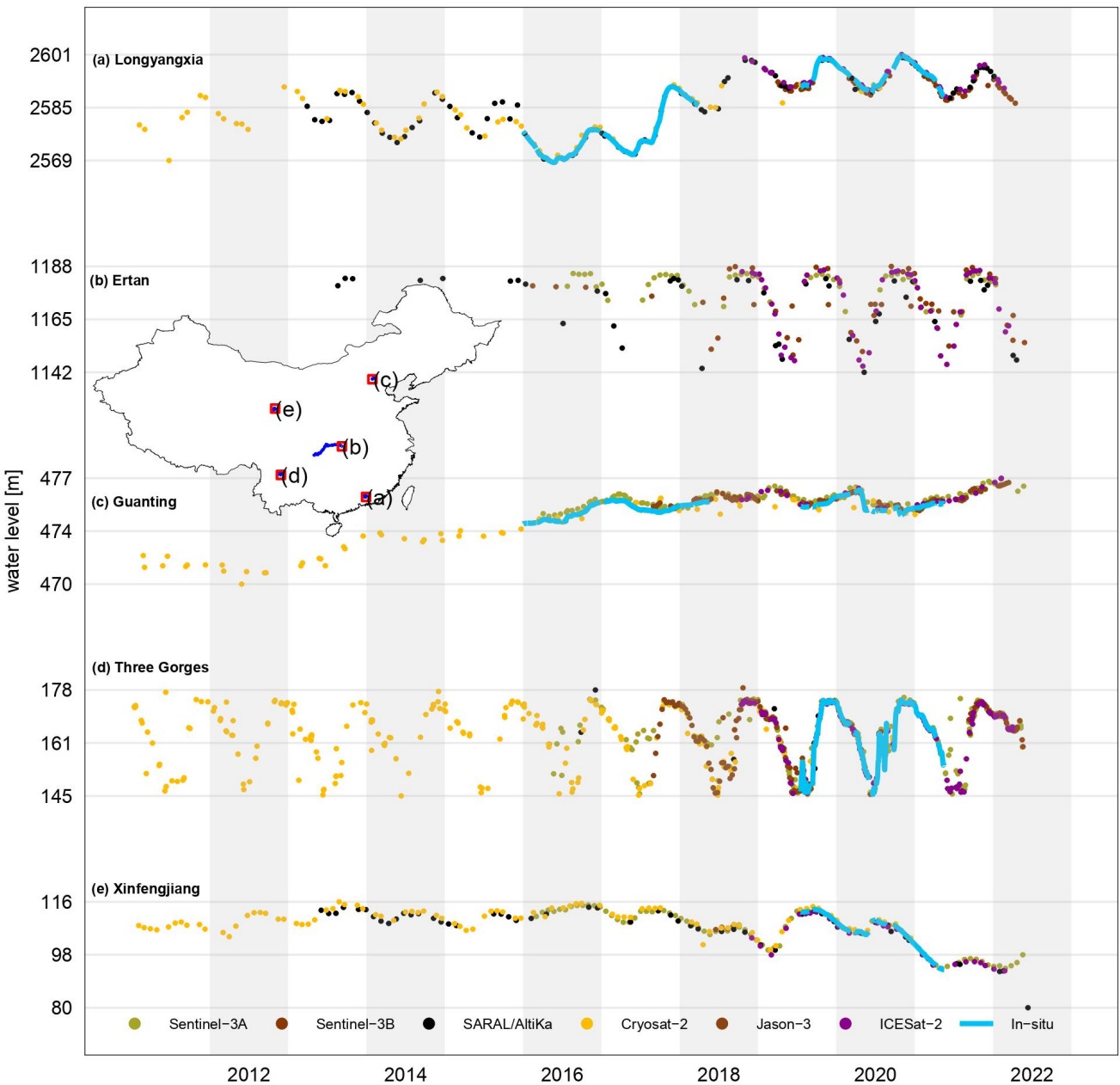

**Figure 5.** High-rate water level time series over reservoirs Xinfengjiang (264 km²) from Sentinel-3A, SARAL/AltiKa, Sentinel-3B, CryoSat-2 and ICESat-2; Three Gorges (852 km²) from Sentinel-3A, SARAL/AltiKa, Sentinel-3B, CryoSat-2, ICESat-2, and Jason-3; Guanting (90 km²) from Jason-3, CryoSat-2, ICESat-2, Sentinel-3A, and Sentinel-3B; Ertan (65 km²) from Jason-3, SARAL/AltiKa, ICESat-2, Sentinel-3A, and Sentinel-3B; and Longyangxia (285 km²) from SARAL/AltiKa, CryoSat-2, ICESat-2, and Sentinel-3B.

### 3.3.2 Reservoir water surface area

Res-CN provides monthly reservoir water surface area data derived from Landsat and Sentinel-2 images during 1984-2021, along with their detailed evaluation reports (see Section of data availability). We compare these datasets with in situ water levels and altimetric measurements as well as other areal datasets (GRSAD and RealSAT). RealSAT generated 681,137 monthly Lake-surface area maps from Landsat images during 1984-2015 using an ORBIT (Ordering-Based Information Transfer) approach that has been validated on 94 large reservoirs. As opposed to RealSAT, which generated new static lake polygons from water occurrence data, GRSAD used existing static surface water polygons, HydroLAKES and GRanD, to create monthly areas for 6,817 global reservoirs based on Landsat images over the last 35 years. The 139 reservoirs with daily in-situ observations generally show good agreement in terms of reservoir area and in situ water level time series with 81% having CC values higher than 0.5. The CC value is expected to decrease for reservoirs with small areal sizes or steep banks. As an example, the CC values of Gutian and Hengjiang reservoirs (10 and 2 km$^2$) are 0.37 and 0.06, respectively. There is also a high median CC value of 0.70 in Res-CN water surface area against altimetric water level time series. As compared to HR altimetric water levels, approximately 63% of 244 compared reservoirs have good CC values exceeding 0.5, including 69 reservoirs with CC values > 0.8. For SR altimetric water levels, approximately 63% of compared 557 reservoirs have good CC values exceeding 0.5, among which 212 reservoirs show very good agreement with CC values > 0.8. To compare Res-CN with these two datasets, we spatially joined reservoir polygons from all datasets, identifying reservoirs with more than 90% overlap. The subset of reservoirs shows good agreement with GRSAD (Fig. 6a, median CC value of 0.65, rBIAS = -10%, rRMSE = 22%, n = 488) and RealSAT (Fig. 6b, median CC value = 0.64, rBIAS = -5%, rRMSE = 20%, n = 288). Since RealSAT and our collected in situ observations do not overlap, we validated only the GRSAD datasets against in situ water levels. GRSAD also showed a good agreement with CC values higher than 0.5 for 58% of 139 reservoirs, including 38 reservoirs that showed a very good agreement with CC values > 0.8. In sum, these comparisons suggest that our data set is reliable.

Uncertainties in surface water area estimates are generally attributed to satellite images and algorithms. As reported by Zhao et al. (2022), the uncertainty of Landsat-based GRSAD areal dataset is 6.1%. In this study, we generated a more reliable reservoir water area product by fusing both Landsat and Sentinel-2 images (Fig. S10), using an algorithm that can largely reducing the impacts of cloud contaminations (Donchyts et al., 2022). There is strong evidence to suggest that this algorithm performs well in this regard, as it has been widely validated in 768 reservoirs of different sizes and climate zones located in Spain, India, South Africa, and the USA (Donchyts et al., 2022). Nevertheless, some limitations and future developments should be considered. Our first option is to use Sentinel-1 data to provide more information in cloudy regions. Furthermore, the algorithm may be improved by either multiclass Otsu or using advanced machine learning methods.

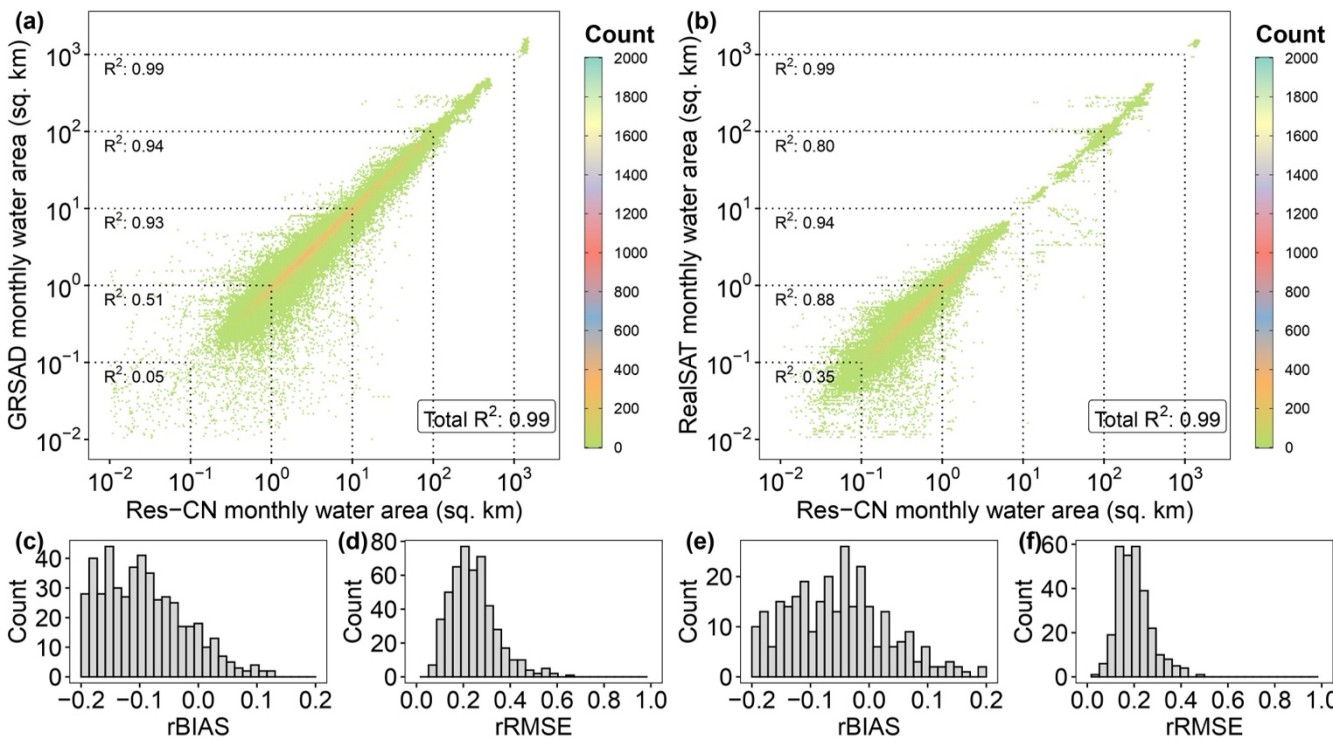

**Figure 6.** Cross validation of reconstructed monthly reservoir area values by comparing to two existing global datasets. (a, c, and d) Comparison between reconstructed monthly reservoir areas and GRSAD. (b, e, and f) Comparison between reconstructed monthly reservoir areas and RealSAT.

### 3.3.3 Reservoir storage anomaly

The Res-CN database provides monthly reservoir water storage anomaly for 3254 Chinese reservoirs during 1984-2021 using DEM's area-storage model, along with their detailed evaluation reports (see Section of data availability). 74% reservoirs (89 of 119 validated reservoirs with in situ data) have good data quality with a RMSE value below 0.2 $km^3$ and a NRMSE value below 30%. NRMSE, CC, and RMSE have median values of 21%, 0.53, and 0.03 $km^3$, respectively. Lowest NRMSE is 7% in the Luhun reservoir, which has a high CC value of 0.90 and low RMSE value of 0.016 $km^3$. Figure 7 shows variations in water storage in small, medium, and large reservoirs located in different climate zones. The remotely sensed storage anomalies generally agree with the observations represented by the statistical metrics, although some large discrepancies occur in peak values. We find that our error statistics in terms of NRMSE are a bit higher than previous works in terms of NRMSE below 20% (Zhong et el., 2020). The errors result from the inaccuracy of the area-storage model developed by DEM as well as the error of water surface areas at certain reservoirs. To solve this problem, we provide another type of storage variation estimates for 335 reservoirs using satellite water surface areas and water levels (see section 2.3, Shen et al., 2022b). The accuracy of storage anomalies is improved, with the median statistics of CC, NRMSE, and RMSE of 0.89, 11%, and 0.021 $km^3$, respectively.

The uncertainties in storage anomalies are primarily attributed to three sources, i.e., the altimetric water level, water surface area estimations from Landsat and Sentinel-2 images, and the error resulting from their combination (the hypsometric curve). Fig. S11 provides an example that illustrates how the uncertainties in satellite datasets propagate to storage anomalies. According to Shen et al. (2022), the primary source of error in storage anomaly is water surface area and the hypsometric curve. Regarding the water surface area, after applying the algorithm developed by Donchyts et al. (2022), these errors and impacts can be reduced to a large extent. Meanwhile, we employed five hypsometric relationships, and the one with the highest $R^2$ value for further use. For more than 80 % reservoirs, the $R^2$ values are greater than 0.5, providing a strong foundation for storage anomaly estimates. Nonetheless, the current satellite sensors have limitations, as evidenced by the significant discrepancies observed in peak values (Figure 7). The increasing temporal resolution and data accuracy of satellite datasets, such as the SWOT mission, will likely improve the accuracy of storage anomaly estimates in the future.

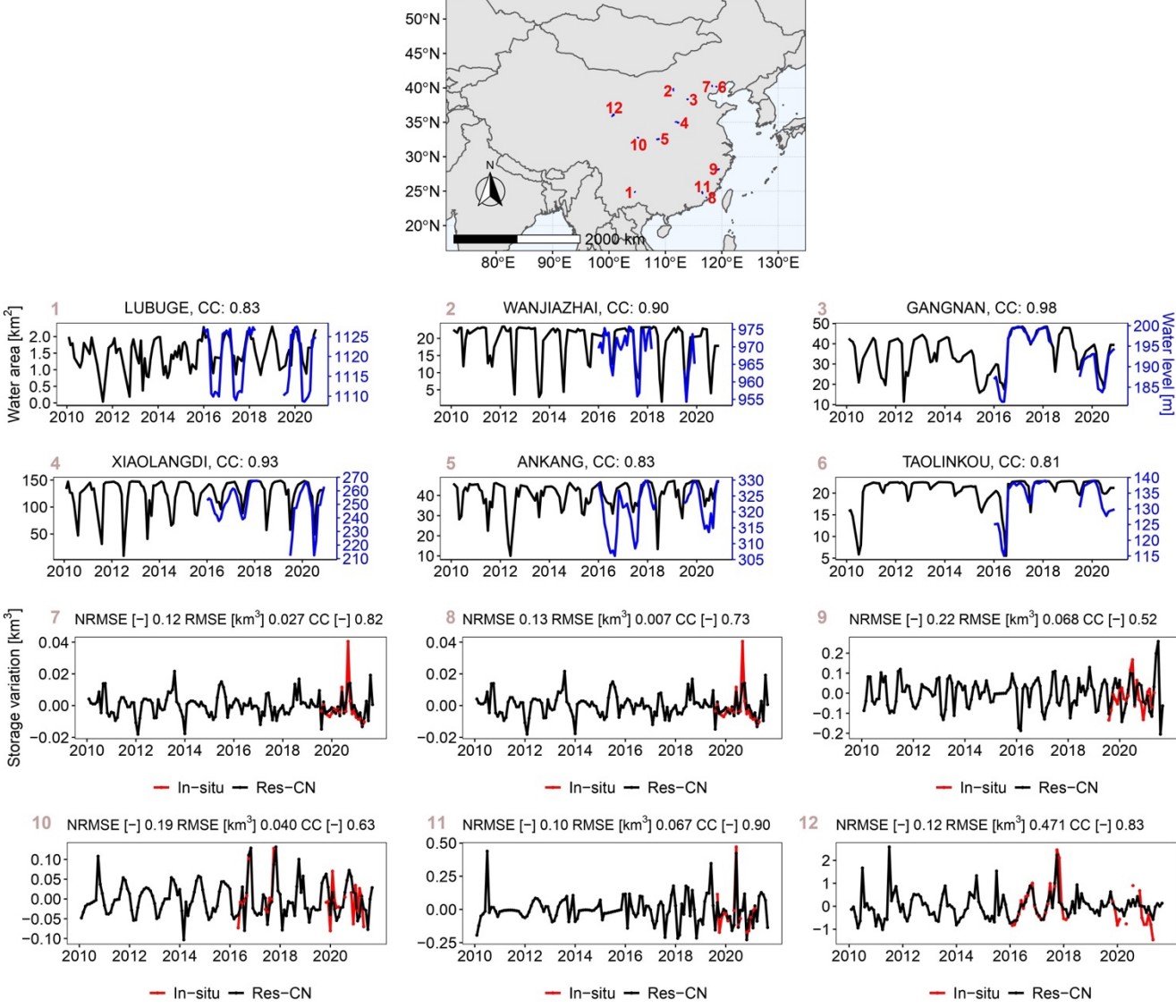

**Figure 7.** Time series of water surface area and storage anomaly in selected reservoirs. RMSE (km³), NRMSE, and CC values are given at the top of each subplot when in situ observations available. Note that: time series of water surface area and storage anomaly of the remaining reservoirs are available in our datasets.

### 3.3.4 Reservoir evaporation

Res-CN provides monthly reservoir evaporation values for 3254 Chinese reservoirs during 1984-2021. Detailed validations of the algorithm can be found in Zhao et al. (2019; 2022) and Tian et al., (2021). The validation of simulated evaporation at an annual scale from Tian et al. (2022) at 47 reservoirs was summarized in Fig. S12 through a literature review. The results in Fig. S12 indicate that the modeled average annual evaporation rates match well with the

observed rates. Specifically, the percent bias (PBIAS), Nash-Sutcliffe efficiency (NSE), and root-mean-square error (RMSE) were found to be 0.02%, 0.82, and 11.2 mm, respectively. This high level of agreement suggests that the Penman method is a reliable approach for calculating reservoir evaporation rates in China. Fig. S13 shows the long-term mean meteorological variables that were used to calculate the evaporation rates. We found that reservoirs located in the southern and coastal areas have significantly larger values than other areas due to larger radiation values. For example, the mean evaporation for 613 reservoirs in Pearl River basin is 1,210 mm/year, while the mean evaporation for 26 reservoirs in Songhua River basin is 717 mm/year (Fig. 8a). With respect to the mean reservoir areas (Fig. 8b), small and medium reservoirs are widely distributed across the nation. The CC values between the mean evaporation rates and the surface shortwave radiation, vapor pressure deficit, mean air temperature, and wind speed are 0.88, 0.84, 0.86, and 0.88, respectively.

Despite the good performance of the algorithm, some limitations are worth noting. Uncertainties in the evaporation estimates are generally attributed to three major sources, i.e., the input meteorological forcings, area estimations from Landsat images, and the limitations of the algorithms. As reported by Zhao et al. (2022), the uncertainty of reanalysis datasets is 7.22%, and the TerraClimate datasets used in this study produce the most reliable evaporation estimates, resulting in a total uncertainty value of 9.93%. Regarding reservoir water surface area, after applying the algorithm developed by Donchyts et al. (2022), these impacts can be reduced to a large extent. There is some room for improving evaporation rate calculation, such as considering the effects of stratification on water temperature or including the advective heat fluxes from reservoir inflow, outflow, and groundwater.

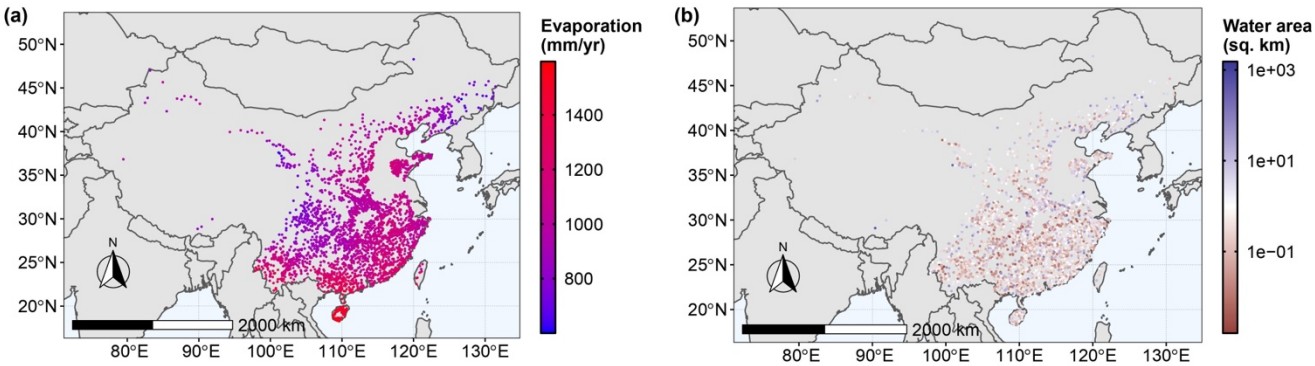

**Figure 8.** Validation of reconstructed monthly reservoir evaporation values. (a) Long-term mean evaporation rates and (b) water surface areas during 1984-2021.

### 3.4 Res-CN products for reservoir catchment-level characteristics

#### 3.4.1 Topographic characteristics

19 topographic attributes are provided in Res-CN (Table S10). The catchment area, slope, and catchment-level

elevation (including mean, maximum, minimum, and standard deviation values) were calculated based on MERIT Hydro and MERIT DEM (Yamazaki et al., 2017; 2019). Slope was calculated using the algorithm developed by Horn (1981). Moreover, we determine 10 indices of catchment shapes and stream network as they are vital in runoff generation and flood situations. Mvert_ang is defined as the angle between the longitudinal axis and the north direction, while mvert_dist is a catchment's longitudinal axis distance. These two indices could determine the relative precipitation trajectory in combination with wind speed. Elongation_ratio is a measure of roundness (i.e., the higher, the rounder) of the catchment, and calculated according to Subramanya (2013). Strm_dens is often used for comparing catchments as it is a function of many catchment attributes such as climate, soil, and geology. We used the MERIT-Hydro database (Yamazaki et al., 2019) to calculate stream density and length within a catchment. The form factor, shape factor, circulatory ratio, relief of each catchment are also provided. Besides, we also added "resArearatio" to describe the proportion of the reservoir water surface area and storage to the catchment area (Fig. S14). High average catchment elevations and slopes are most apparent in the western China, which extend from western Yunnan–Guizhou Plateau and southern Qinghai–Tibet Plateau to the northwestern areas (Fig. 9a and b). The streamflow density and resArearatio are relatively high in the central of the Yangtze River basin, where the artificial reservoirs are densely distributed (Fig. 9c and d). High elongation ratios are widespread in China and mvert_ang shows high values in the south (Fig. 9e and f).

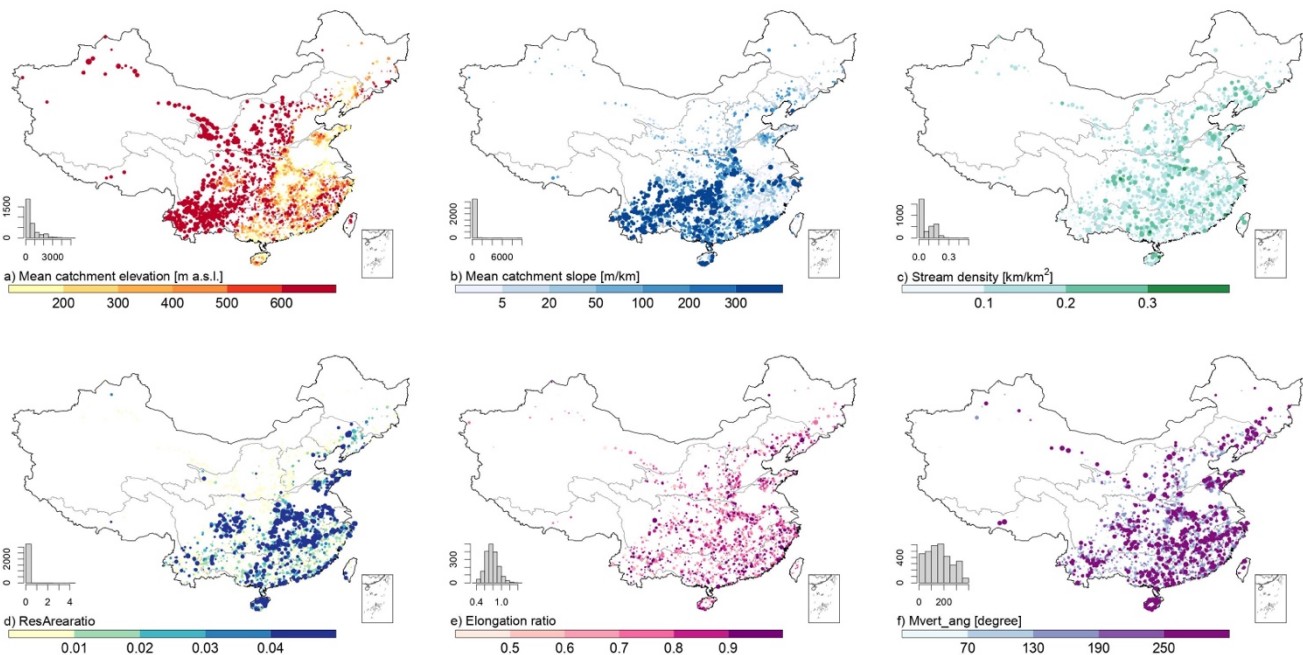

**Figure 9.** Distribution of selected topographic characteristics of intermediate catchments and full catchments. Each category's histogram indicates the number of basins (out of 3254). In a histogram, the X-axis represents the number of basins, while the Y-axis represents each subplot's title. Circle sizes are proportional to catchment areas.

### 3.4.2 Climatic characteristics

Res-CN includes daily meteorological time series and 11 attributes reflecting aspects of climatic characteristics (Tables S11-12). In this study, we used the National Station-based Climatic Data set V3 (NSCD) to compute the catchment-level climatic characteristics at full catchments and intermediate catchments. NSCD provides daily meteorological time series during 1951–2020 and has near 800 stations in China, with the longest period of gauged observations of precipitation, temperature, evaporation, wind speed, pressure, sunshine duration, relative humidity, and near surface temperature. The ground sites were sparse before 1970 (Hao et al., 2021). To ensure data quality and match the periods of other datasets in Res-CN, we used the latter 41 years (from 1980 to 2020) to generate a gridded data set based on the inverse distance weighting interpolation technique (Fig. A2). Figure 10 shows the spatial distribution of selected long-term mean meteorological forcings of full catchments and intermediate catchments. The precipitation, and mean air temperature have an increasing trend from northwest to southeast. The sunshine durations are high in the northern areas where latitude is high, and the maximum value is 8.7 hour/day. Wind speed is generally high in the northern mountainous areas, southwestern, and coastal parts. The average evaporation is positively correlated with sunshine duration and is higher in Yellow River basin and southwest parts.

We calculated nine attributes for NSCD based on meteorological data between 1 October 1990 and 30 September 2020 to reflect aspects of climatic characteristics. Using the Global Aridity Index and Potential Evapotranspiration Climate Database version 3(Zomer and Trabucco, 2022), we derived the reference evapotranspiration ($ET_0$) and aridity index. Aridity is often calculated as a function of precipitation, temperature and $ET_0$, and quantifies the precipitation availability for atmospheric water demand. Long-term daily precipitation, reference evapotranspiration, and aridity index (Fig. 10f) characterize the long-term climatic characteristics. The seasonality of precipitation (Fig. 10g) and the fraction of precipitation falling as snow (not shown) are two attributes characterizing seasonality, which yield the yearly maps of sinusoidal precipitation cycle. Short-term events (e.g., heavy rainfall and drought) are characterized by the frequency and duration of heavy precipitation/dry days, and their most likely seasons of occurrence (Fig. 10h, i). High precipitation is most likely to occur in summer (Fig, 10h) for 86 % of all 3254 catchments, whereas dry days usually occur in winter (Fig, 10i) for 56 % of them.

One key limitation of the NSCD is that these gauged stations are unevenly distributed across the nation, and densely grouped in the eastern and middle parts, which may affect the accuracy for some catchments. Nevertheless, the NSCD meteorological dataset has been widely used as the most reliable observational reference by many studies (Gu et al., 2022). It is produced from a larger number of gauged stations and followed by strict quality-assurance procedures and consistence-check. Its accuracy and completeness of each meteorological variable from 1951 to 2020 are significantly improved compared with similar data products in China, the missing rate of data of each element is generally below 1%, and the correct rate of data is close to 100%. Relative humidity suspicious rate 0.6%, large evaporation suspicious rate 1.2%, great wind suspicious rate 1.6%, the suspicious rate of other elements does not exceed one thousandth (Hao

et al., 2021). The error rate of sunshine hours is one ten thousandth, and there is no wrong data for other meteorological variables.

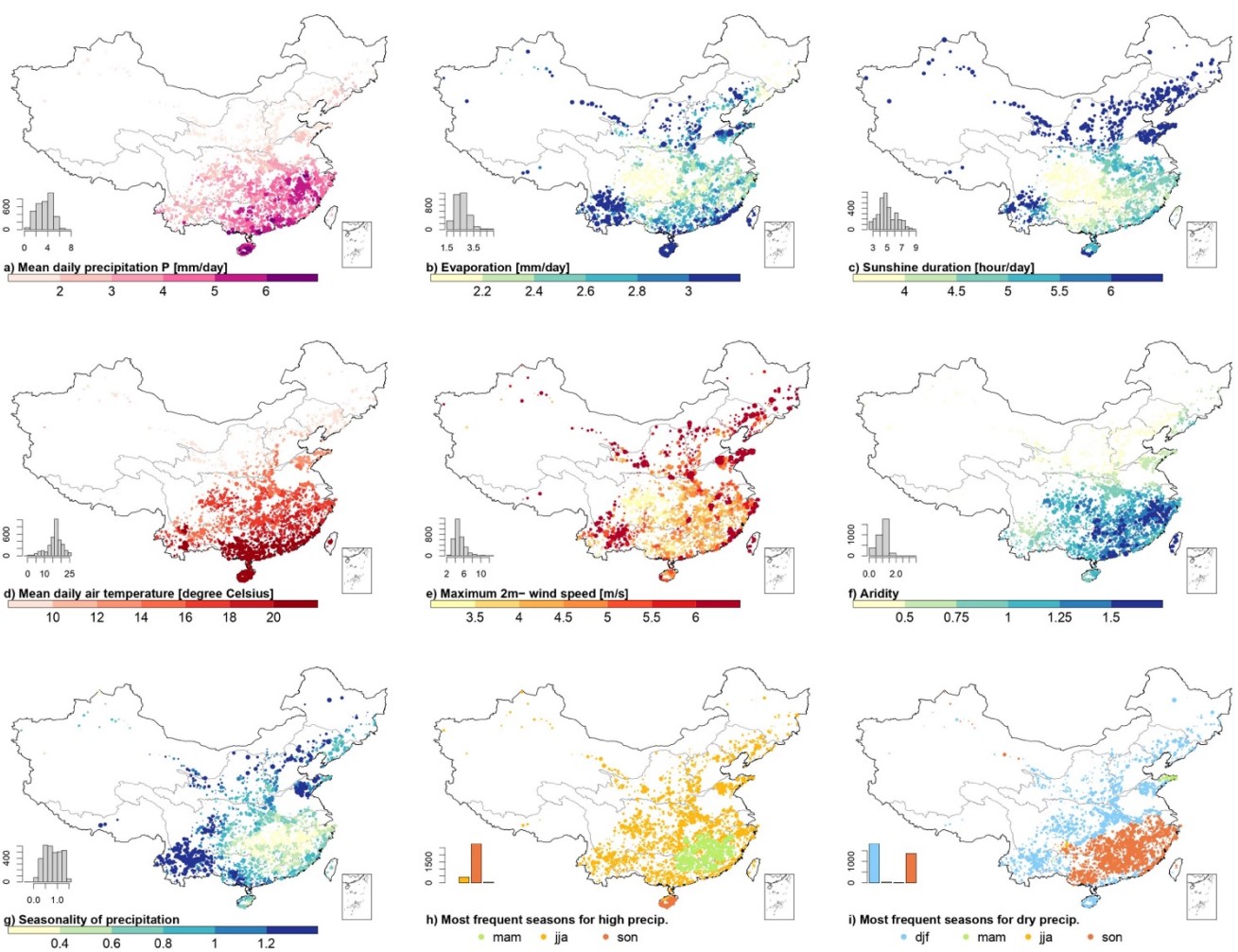

**Figure 10.** Distribution of selected long-term mean meteorological variables and climate indices of intermediate catchments and full catchments. Each category's histogram indicates the number of basins (out of 3254). In a histogram, the X-axis represents the number of basins, while the Y-axis represents each subplot's title. Circle sizes are proportional to catchment areas.

### 3.4.3 Land cover characteristics

We provide three vegetation indices (EVI, enhanced vegetation index; LAI, leaf area index; NDVI, normalized difference vegetation index), two indicators of land use (GPP, gross primary productivity; NPP, net primary production), two average rooting depths (50 % and 99 %), and 10 land cover classes in each catchment (Table S13). LAI was derived from the MODIS MCD15A3H dataset with a temporal resolution of 4 days and a spatial resolution of 500 m (Myneni et al., 2015) and used for characterizing vegetation growth. NDVI and EVI were derived from the MODIS MOD13Q1 (Didan,

2021) and used to monitor and classify vegetation. These vegetation indices were computed as the maximum, minimum, or difference (e.g., $LAI_{max}$, $LAI_{min}$, and $LAI_{diff}$). $LAI_{max}$ measures the maximum evaporative and vegetation interception

capacity, and $LAI_{diff}$ shows its temporal variations. GPP and NPP were derived from MODIS MOD17A2H (Running et al., 2021a) and MOD17A3HGF (Running et al., 2021b) dataset, respectively, and calculated for the whole period of 1 February 2000 to 1 January 2022. Meanwhile, the time series of both vegetation indices and indicators of land use are also provided in the Res-CN. Rooting depths are important parameters to characterize the water holding capacity underground and annual evapotranspiration of topsoil. We calculated two average rooting depths (i.e., 50 % and 99 %)

based on the IGBP (International geosphere–biosphere programme) classification (Zeng 2001).

Each catchment was described using 10 land cover classes based on ESA WorldCover 10 m (Zanaga et al., 2021). This dataset is a new baseline of global land cover product at 10 m spatial resolution for 2020 in almost near-real time based on both Sentinel-1 and Sentinel-2 data. Sentinel-1 can provide complimentary information on the observed structural characteristics of land cover in areas where the Sentinel-2 images were covered by clouds. Thus, the combination of

Sentinel-1 and Sentinel-2 data enables mapping land cover almost in real time. It includes 11 land cover classes and an overall accuracy of 75%, providing valuable information for food security, carbon assessment, biodiversity, and climate modelling. The dominant class and fractions of each class were computed at the GEE platform.

Some limitations of these datasets are identified. First, misclassification of ESA WorldCover occurs in areas of irrigated agriculture and wetlands due to the high similarity of their hyperspectral spectrum. Second, although Res-CN provide

time series of vegetation indices, it should be noted that NDVI often provides inaccurate measurements of vegetation density, the accuracy of which can only be guaranteed by long-term measurements. In addition, NDVI cannot provide quantitative estimates of vegetation density, so other attributes (i.e., LAI) are provided as a complement.

Trees are prevalent across the nation, and grasslands and croplands are another two dominant land cover in China. Grassland has higher coverage largely in the Hai River Basin and the Yellow River Basin (Fig. 11c). Croplands are

widespread in China, especially in the Yellow River, Yangtze River, and Huai River basins with a low mean slope (Fig. 11d). Fraction of barren sparse vegetation is quite small across the nation (Fig. 11e). Natural wetlands or water bodies are mainly distributed in the Yangtze River Basin and areas surrounding the Bo seas, and these water bodies are mainly artificial reservoirs and natural lakes (Fig. 11f). Catchments with a relatively high faction of snow and ice are mainly located in the Tibet plateau, the source region of large rivers in China (not shown). A small proportion of the catchment

area is typically considered as "built-up", and 6% of the catchments have impervious area greater than 0.05 (not shown). There is a spatial correlation between $LAI_{max}$ and $LAI_{diff}$ with trees fraction (CC = 0.74 and CC = 0.60, respectively). $LAI_{diff}$ has values similar to $LAI_{max}$ over most areas but should be smaller in areas with a high proportion of trees due to permanent green cover. Negative correlation is evident between the $NDVI_{max}$ (Fig. 11i, CC = −0.78) and the mean catchment elevation (Fig. 9c).

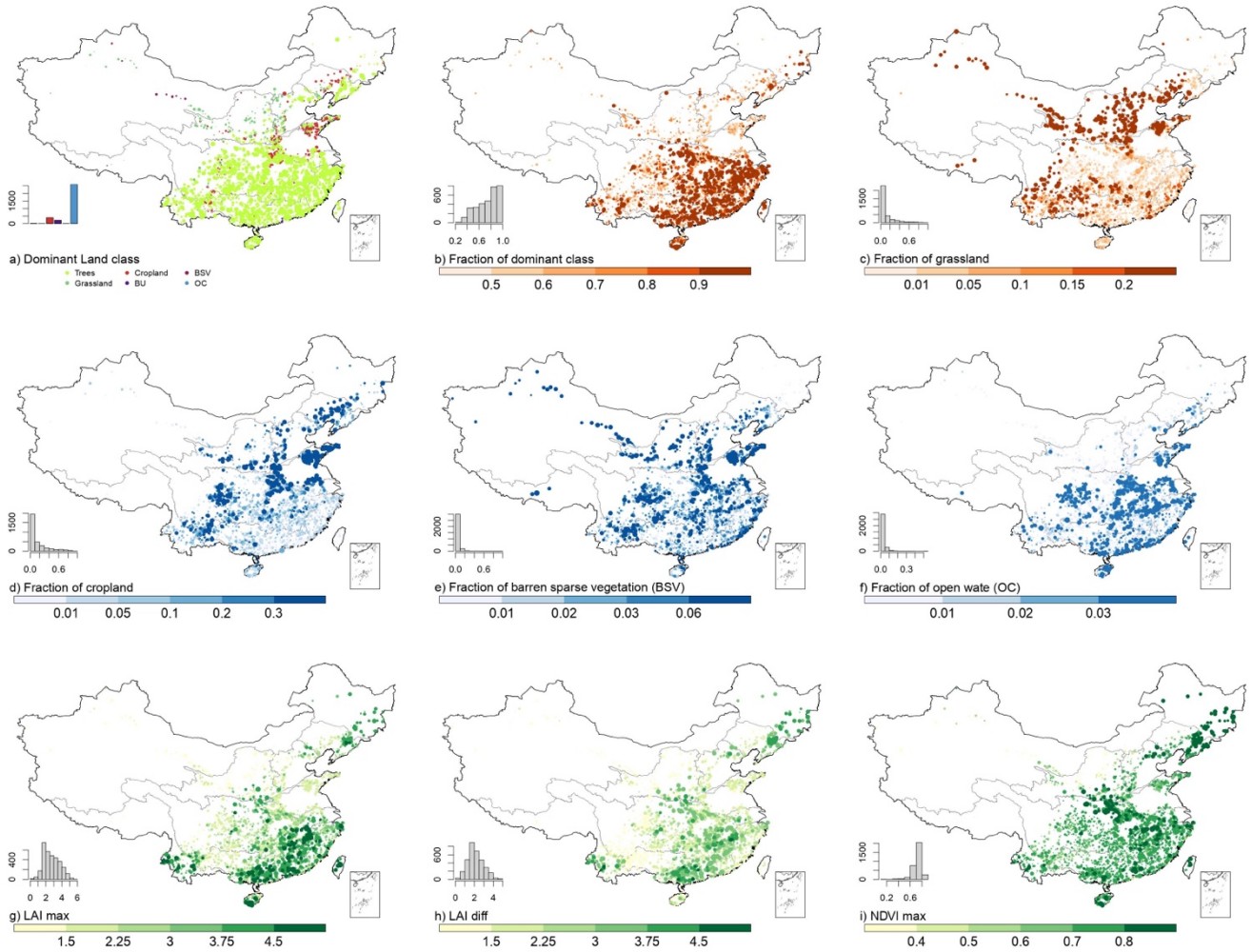


**Figure 11.** Distribution of selected land cover characteristics of intermediate catchments and full catchments. Each category's histogram indicates the number of basins (out of 3254). In a histogram, the X-axis represents the number of basins, while the Y-axis represents each subplot's title. Circle sizes are proportional to catchment areas.

### 3.4.4 Soil & geology characteristics

Res-CN provided 154 attributes to characterize physical and chemical properties of soil (Tables S14). The pH in $H_2O$, cation exchange capacity (CEC), bulk density, and organic carbon content were determined from the SoilGrids250m dataset (SG250, Hengl et al., 2017). The SoilGrids250 dataset predicted soil properties at six different soil layers (i.e., 0- 0.05m, 0.05-0.15m, 0.15-0.3m, 0.3-0.6m, 0.6-1m, and 1-2m) using machine learning techniques, utilizing data from approximately 150,000 soil profiles and 158 environmental covariates derived from remote sensing data on a global

scale. Data within Res-CN are provided for each soil layer and all soil layers using depth-weighted averaging method. Soil pH is an important variable as it controls many other soil biological, chemical, and physical properties. CEC defines

the sum of exchangeable cations that soil can hold and is therefore a measure of fertility nutrient retention capacity. The density and organic carbon content refer to the mass of organic carbon per unit volume and mass, respectively. Besides SG250, the dataset of soil hydraulic and thermal parameters produced by Dai et al. (2019) was also used. Dai

et al. (2019) generated six soil layers of global soil hydraulic and thermal parameters with four products of vertical profiles available using multiple PTFs (Pedotransfer Functions) based on SG250 and soil dataset from Shangguan et al. (2014). We adopted the products with the vertical resolutions of SG250 and computed soil characteristics (saturated water content, saturated hydraulic conductivity, and other thermal parameters) for all soil layers. Additional attributes derived from Shangguan et al. (2013) include clay, silt, sand, and rock fragment proportions, soil profile depth, and soil

organic carbon content. With the use of the polygon linkage method, this database provides soil physical and chemical attributes for land surface modeling in China at a 30 arcsec resolution. Proportions of clay, silt, sand refer to the fractions of the particles < 0.002 mm, particles ≥ 0.002 mm and ≤ 0.05 mm, and particles > 0.05 mm and < 2 mm in the fraction of particles smaller than 2 mm, respectively. SOC is a key variable for ecosystems and affects moisture regimes and ground thermal.

Geology characteristics are described by 19 attributes (i.e., subsurface porosity, permeability, and lithological classes) derived from global lithological map (GliM, Hartmann and Moosdorf, 2012) and global hydrogeology maps, (GLHYMPS, Gleeson et al., 2014) datasets (Table S15). The two important parameters for ground water modeling, subsurface porosity and permeability, came from the GLHYMPS. Subsurface porosity is a measure of the ability of the subsurface to store water, while permeability is a measure of the ability of the rock to transmit fluids. Both these two parameters

show a high spatial correlation with GliM map, as hydraulic properties in GLHYMPS are based primarily on lithological classes of GLiM. We computed the catchment-level characteristics by applying an arithmetic mean method for porosity and arithmic scale geometric mean method for permeability. The lithological classes were derived from the GliM, which was created by summarizing 92 regional lithological maps and offers three classification levels of detail. In this study, we adopted the first level of GLiM, that has 16 lithological classes. Proportions of each lithological class and the

dominant class are documented in our datasets.

There are some limitations with these datasets. Firstly, The GLHYMPS module is primarily useful for analyzing regional scales, i.e., scales larger than 5 km, where the effects of local heterogeneities such as fault zones are negligible (Gleeson et al., 2014). Secondly, GLHYMPS is not adequate for analysis at regions dominated by unsaturated processes such as deeply weathered soils, as it is modeled for saturated conditions (Huscroft et al., 2018). Thirdly, data quality varies

depending on location, based on raw regional geological maps available in different resolutions and data quality. In this study, the resolution of the Chinese raw data sources is slightly lower than that of GLiM (Hao et al., 2021).

The soil pH value is high in the northern and northeastern China, and the saturated water content is low in this area (Fig. 12a and b). High values of CEC can be found in central and northeastern China and forested parts on the Qinghai–Tibet Plateau (Fig. 12c). The clay content is low in the northern China, while the sand content shows the opposite

pattern (Fig. 12d and e). The silt content has a wide predominance pattern in China, particularly in the middle and northeastern parts (Fig. 12f). Moreover, we found that soil characteristics are correlated with other attributes from Res-CN as they are predicted by fusing multisource covariates such as climate and landscape attributes. For example, the soil texture (i.e., sand, silt, and clay contents) shows similar pattern of the aridity index (Fig. 12e and Fig. 10f). SOC has a high correlation value with surface slope (Fig. 12c and Fig. 10b, $R^2$ = 0.94). The geology attributes have a clear

latitude distribution, and the main lithological classes include acid plutonic rocks (21% of the catchments), siliciclastic sedimentary (20%), mixed sedimentary (19%), and carbonate sedimentary (17%). Medium subsurface porosity and high permeability are typically in north and coastal surrounding the Bohai Sea areas with abundant unconsolidated sediments, and high porosity and low permeability are often associated with mixed sedimentary rocks in northern Inner Mongolia, western Yunnan–Guizhou Plateau and southern Qinghai–Tibet Plateau (Fig. 12h, i, and e). Interestingly,

high values of subsurface porosity are not necessarily accompanied by high values of permeability, producing a more heterogeneous spatial pattern (Fig. 12h) than that in Fig. 12i. There may be differences in permeability and porosity due to the different rock structures of GLiM (Gleeson et al., 2014).

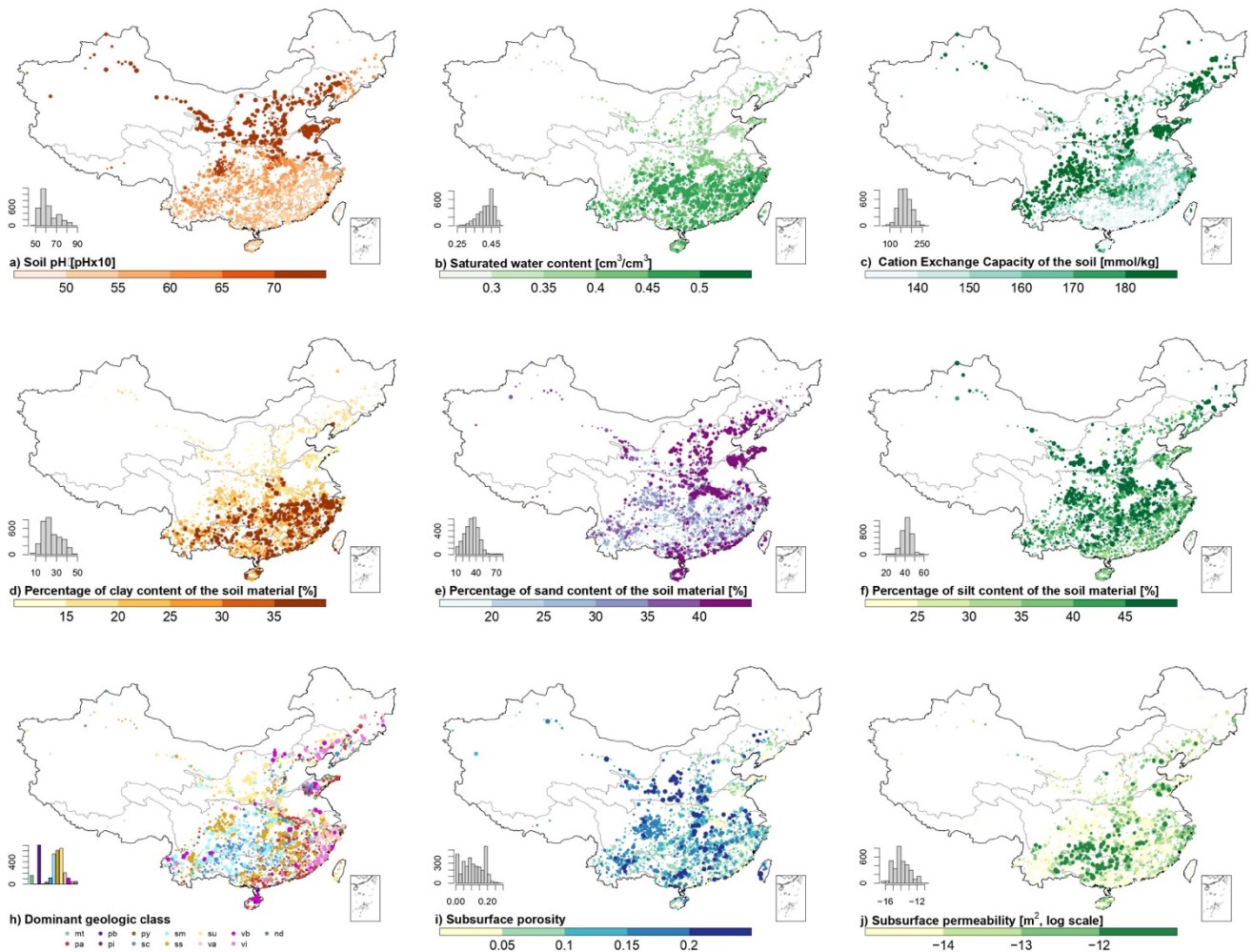

**Figure 12.** Distribution of selected soil and geology characteristics of intermediate catchments and full catchments. Each category's histogram indicates the number of basins (out of 3254). In a histogram, the X-axis represents the number of basins, while the Y-axis represents each subplot's title. Circle sizes are proportional to catchment areas.

### 3.4.5 Anthropogenic activity characteristics

We computed four categories of anthropogenic activities in the catchments, including population count, human footprint, nighttime lights, and road density (Table S16). Population counts were derived from the GPW v4.11 (Gridded population of the world, Center for International Earth Science Information Network – CIESIN - Columbia University, 2018). This collection produces human population estimates for five years (i.e., 2000, 2005, 2010, 2015, and 2020) at a gridded resolution of 30 arc-seconds. Human footprint is a measure of human activities that use natural resources on Earth and was extracted from the Global Human Footprint dataset (Venter et al., 2016). This database measures the cumulative environmental impact of indirect and direct human activities in 1993 and 2009, and was produced by eight

inputs: electric infrastructure, population density, built environments, pastures, crop lands, roads, railroads, and navigable waterways. Nighttime lights are a measure of human activity intensity and were derived from the DMSP-OLS Nighttime Lights version 4 (Defense Meteorological Program-Operational Line-Scan, Doll, 2008). This database can detect visible and near-infrared emission sources at night and consists of cloud-free composites made using all the available archived DMSP-OLS smooth resolution data at a resolution of 30 arc-seconds. The "avg_lights_x_pc" in this

dataset is used in this study and represents the mean value of cloud-free light detections in the visible band. The road density was obtained from Global Roads Inventory Project (Meijer et al., 2018), providing global raster datasets of road density at a 5 arcminutes spatial resolution. Using 60 geospatial datasets on road infrastructure, this inventory gathers, harmonizes, and integrates over 21 million km of roads by country.

These datasets are subject to some limitations. First, cumulative pressures of human footprint are static through time

due to a lack of available data, which would lead to an underestimation of human footprint if those pressures expanded at an above-average rate. Second, some static pressures, like the pollution and invasive species are not considered in the cumulative pressures of human footprint. Third, the GRIP datasets cannot quantify historic road expansion due to the missing information on the year of construction. Fourth, DMSP-OLS has the blooming effect (i.e., overestimation of lit area) due to the low spatial resolution and the reflectance of light from adjacent water bodies.

Figure 13 illustrates the spatial distribution of four anthropogenic indices in the catchments. Road density, population count, lights, and human footprint show the similar patterns, suggesting that intense human activities are distributed in the coastal lines surrounding the East and Bo seas, middle, and northeastern China and there is almost no human activity in the northwestern China due to high elevation and harsh environment.

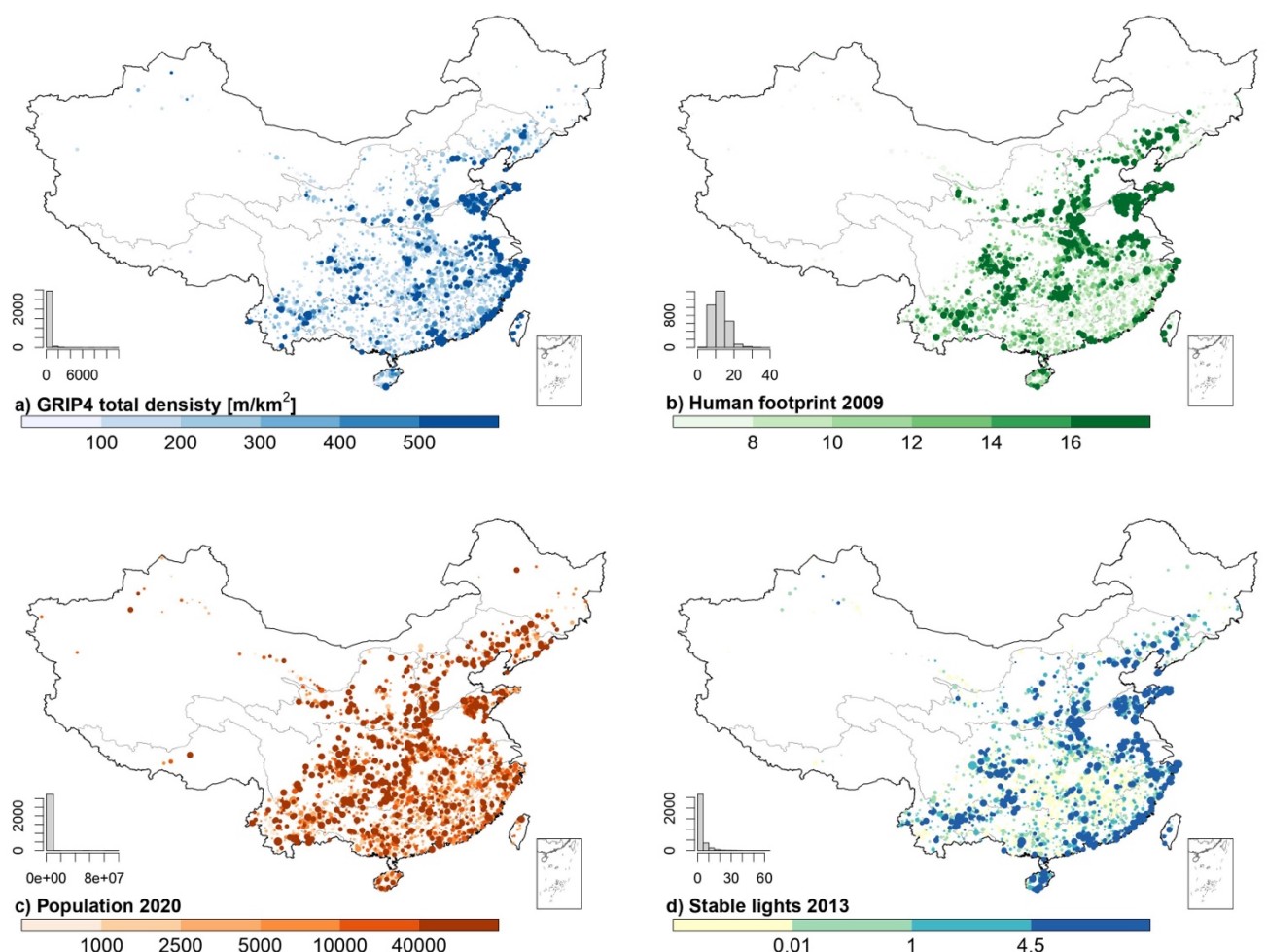

**Figure 13.** Distribution of human activity characteristics of intermediate and full catchments. Each category's histogram indicates the number of basins (out of 3254). In a histogram, the X-axis represents the number of basins, while the Y-axis represents each subplot's title. Circle sizes are proportional to catchment areas.

## 4 Data availability

Res-CN archive can be found here https://doi.org/10.5281/zenodo.7664489 (Shen et al., 2022a). They are distributed with a CC-BY license. The files provided are (A) shapefiles, (B) full catchments containing all catchment-level attributes such as climate, topographic, land cover, soil, geological, anthropogenic activities, and time series of meteorological variables, (C) intermediate catchments, (D) reservoir states (i.e., water level, water surface area, evaporation, and storage anomaly), (E) info English file containing more information of RES-CN, and validation figures containing the

figures depicting the time series of reservoir states and detailed evaluation reports based on in situ data when available. All input datasets are summarized in the supplements and kindly acknowledged.

## 5 Summary, applications and outlook

In this study, we have produced a comprehensive and extensive data of reservoir-catchment characteristics in China, Res-CN, which complementarily improved the existing reservoir datasets. We demonstrated that the construction of Res-CN involved a first known effort to construct catchment-level characteristics of reservoirs for our delineated full catchments and intermediate catchments of reservoirs. 512 static catchment-level attributes for 3254 reservoirs in six categories (i.e., reservoir and catchment body characteristics, topography, climate, soil and geology, land cover and use, and anthropogenic activity characteristics) are included in Res-CN. Additionally, 15 climatic variables were extracted at daily scale, which can drive machine learning models or hydrological models for simulations. Alongside the catchment-level attributes, we produced a significantly enhanced spatial and temporal coverage (e.g., 67% increase in spatial resolution of water level and 225% increase in storage anomaly) of water level (data available for 20% of 3,254 reservoirs), water surface area (99%), storage anomaly (92%), and evaporation (98%) by utilizing multiple satellites such as operational satellite altimeters and imagery data. In situ data of 138 reservoirs are employed in this study as a valuable reference for evaluation, thus enhancing our confidence in the data quality and enhancing our understanding of accuracy of current satellite datasets. We have considered and discussed the deficits, limitations, and uncertainties of Res-CN for further applications.

We envision that Res-CN with its comprehensive and extensive attributes can provide strong supports to a wide range of applications and disciplines. Firstly, our two types of catchments along with their catchment-level attributes allow investigations within individual catchments and interconnected river networks. For example, as illustrated in Figure 2, users may quantify the relative contributions of upstream reservoirs and local drainage catchment on water quality (e.g., algae contributions and water color) of downstream reservoir by tracking temperature and nutrient flows from upstream reservoirs and intermediate catchments (e.g., Hou et al., 2022; Yang et al., 2022). Besides, water and sediment transfer can be also more accurately simulated in such a spatially explicit context if appropriate approaches are used. Machine-learning methods make it possible to predict reservoir storage change at 1- to 3-month lead from reservoir upstream attributes and time-series of reservoir states (Tiwari et al., 2019). Secondly, Res-CN provide thus far the most comprehensive reservoir states in China for assessing impacts of reservoir regulation and dynamics. Tracking the spatiotemporal balance of reservoir evaporative and water storage can provide a basis for local water management in a warming climate (Di Baldassarre et al., 2019). The reservoir operational rules or impacts of reservoir regulation on flow regimes are possibly to be inferred from reservoir water dynamics in Res-CN (Vu et al., 2022). This is particularly true if the reservoir inflow is also utilized. Recently, the gridded natural runoff provided by Gou et al. (2021) provides

exciting opportunities for quantifying the human water regulation in combination with Res-CN (Dang et al., 2022; Shin et al., 2020). Thirdly, our extracted catchment-level attributes can contribute to a better understanding of reservoir water amount and water quality changes by spatially incorporating geophysical and anthropogenic characteristics of their upstream catchments and their respective contributions. For example, cropland in reservoir upstream catchments controls the nutrient-driven primary production, while wetland coverage affects dissolved organic material transport downstream, ultimately impacting primary production and CO2 emissions in lakes (Balmer and Downing, 2011; Borges et al., 2022; Maberly et al., 2013). Gradient and altitude in the reservoir geological attributes may affect greenhouse gas emissions and biogeochemistry of a reservoir (Casas-Ruiz et al., 2020). Furthermore, these catchment-level attributes can be used to explore water fluxes and sediment transportation even in reservoirs that have not been sampled. Studies on cascading patterns in reservoir attributes found that each attribute may display linear function of catchment area, concluding that cascading patterns of each attribute have different implications for dam management (Faucheux et al., 2022). For instance, one study combined knowledge of catchment attributes with economic, climate, and landscape data to inform reservoir removal decisions in California's Central Valley basin (Null et al., 2014). Lastly, carbon dioxide emissions from reservoirs show significant spatial and seasonal variation, highlighting the importance of hydrology in terrestrial–reservoir carbon transfers and the need to consider this effect when plumbing terrestrial carbon budgets. Res-CN also offers exciting opportunities to address changes in reservoir storage that may be linked to carbon dioxide emissions changes.

Although Res-CN presents significant improvements over existing datasets and holds potential for various applications identified above, a few limitations should be acknowledged. Res-CN is generated using GeoDAR v1 shapefiles (Wang et al., 2022) instead of the newly produced datasets by Song et al. (2022), which added an additional near sixty thousand very small reservoir shapefiles (< 1 km$^2$). As this study aims to provide a comprehensive and extensive dataset of reservoir-catchment characteristics in China for a better understanding of reservoir impacts on hydrological and biochemical cycles, these thousands of very small reservoirs are not included in our study. Meanwhile, it is currently not feasible to generate satellite-based datasets for these small reservoirs due to the limitations of current satellite altimetry missions, which are unable to detect such reservoirs because of the sparsity of their altimetric ground tracks. These additional small reservoirs only account for 8% of total water capacity in China. Nonetheless, users can freely access our codes to calculate any reservoir attributes for individual applications, other areas, and can enrich the inventory if new data available.

## 6 Code availability

All scripts for generating our reservoir datasets are available in the data product.

**Supplements.**

Supplement contains many Tables, Figures, and Texts illustrating the methodologies and source datasets for generating the Res-CN, as well as detailed evaluations of our Res-CN datasets and all necessary explanations related to this article.

**Author contributions.**

Y.S. and D.Y. designed the research and initiated the investigation. Y.S. processed the datasets and created the figures. K.N. extracted data from satellite altimeters. M.R. and D.L. reviewed the manuscript. Y.S. wrote and prepared the manuscript with suggestions from co-authors.

**Competing interests.**

The authors declare that they have no conflict of interest.

**Acknowledgements.**

Data from various data centers were acknowledged by the authors. For more acknowledgements, please refer to our supplementary and our datasets. Youjiang Shen also acknowledge the contributions from the EOForChina project.

**Financial supports.**

This study was financially supported by the funding JSPS KAKENHI 21H05002. The first author is supported by the China Scholarship Council.

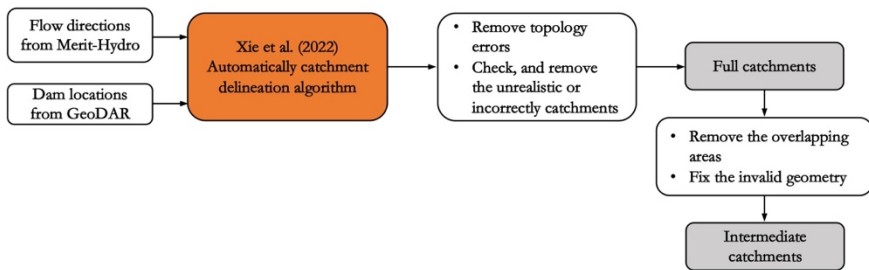

**Figure A1.** Illustration of the datasets provided in our Res-CN.

**Figure A2.** Flowchart of the algorithm for generating reservoir upstream catchments.

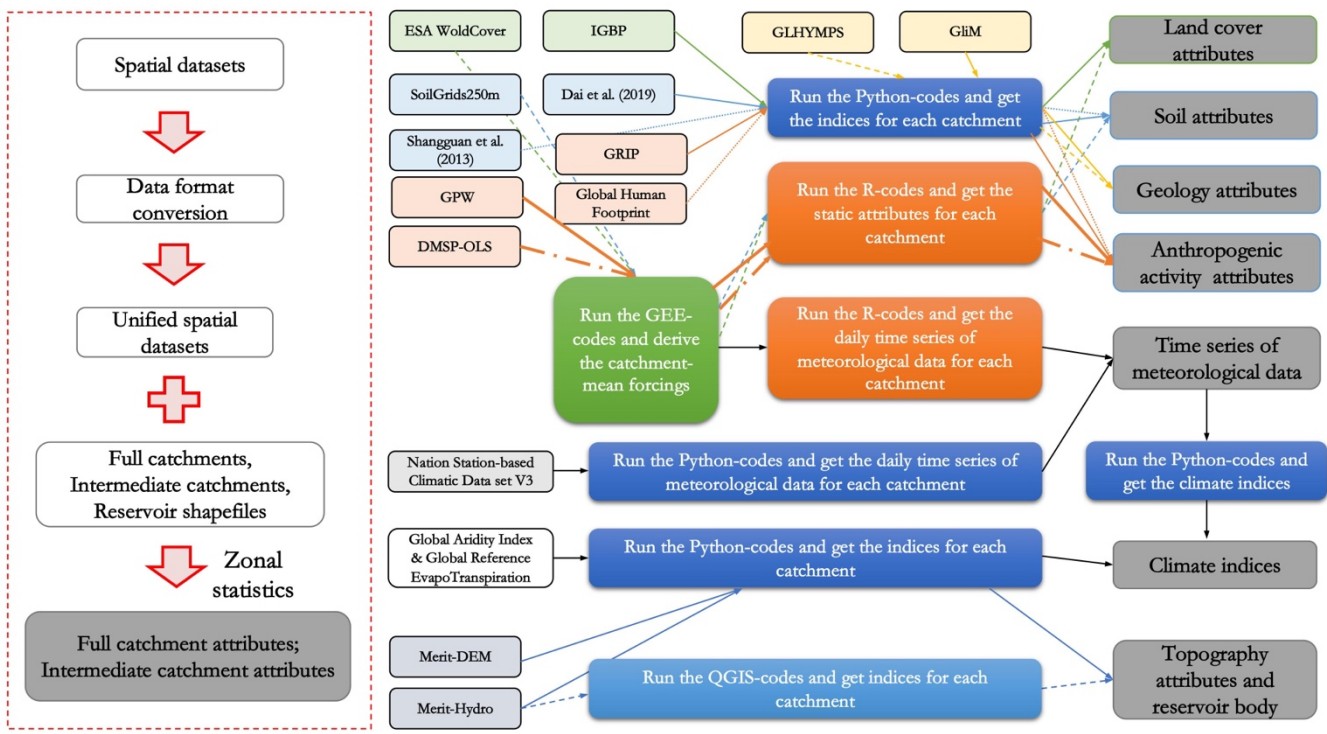


**Figure A3.** Flowchart for generating reservoir upstream catchment-level characteristics. (Overview of the methodology at the left panel, we give the detailed steps for generating each attribute based on our provided codes. Users can freely access our codes in our Res-CN product.

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
