# Peer review of "Res-CN: hydrometeorological time series and landscape attributes across 3254 Chinese reservoirs"

_Earth System Science Data, 2022_

## Community Comment (CC1)

**Reviewer #1 Comment on essd-2022-422 (Anonymous Referee #1)**

Dear Anonymous Referee #1,

We have carefully reviewed your comments and have made the necessary revisions to our manuscript. Please find attached a point-by-point response to your feedback, marked in purple. We hope that our revised manuscript (in red) can help the readers to better understand our study.

Kind regards.

**General comments**

Shen et al. presented a very comprehensive reservoir dataset for China, Res-CN. The new dataset includes water area, water level, storage variations, and corresponding catchment characteristics that derived from multiple sources (i.e., satellite, reanalysis, and observation, etc). The authors also validated Res-CN with in-situ observations at selected reservoirs to demonstrate the accuracy of the dataset. It provides valuable information for hydrological modelers to investigate water managements and the impacts on (eco)hydrological cycle. Although I think Res-CN represents a significant contribution to improve our understanding of reservoir dynamics and water management in hydrological modeling, some parts were not clearly presented /explained in the main text because Res-CN contains extensive information. Additionally, some figures were missing in the supplementary materials. So, I recommend revision before publication. Please find my comments in the following.

R1C0: We thank you for your thoughtful review of our manuscript and for recognizing the comprehensive nature of our Res-CN dataset for China. We are pleased to hear that you agree that our dataset, which includes a variety of data sources such as satellite, reanalysis, and observation, can provide valuable information for hydrological modelers to better understand water management and the (eco)hydrological cycle. We appreciate your feedback on the presentation of our work and understand that the extensive information included in Res-CN may have led to some parts being less clearly explained in the main text and supplementary. We have carefully reviewed your comments and made the necessary revisions to improve the clarity of our manuscript. We have also corrected some mistakes in the supplementary materials.

We thank you for the opportunity to revise our manuscript before publication, and we hope that our updated version meets the standards of the journal. We hope that our Res-CN dataset can serve as an important resource for hydrological modelers and researchers in the field, and we appreciate your time and effort in reviewing our work. If you have any further suggestions or comments, please do not hesitate to let us know.

**Major Comments**

1. The authors mentioned in the introduction Line 99 that in-situ data of 138 reservoirs were used to validate the Res-CN, but the validations at a few reservoirs are shown in the result section, with summary in the main text. It is necessary to show the validations explicitly for all the 138 reservoirs that demonstrate the accuracy of Res-CN.

R1C1: We apologize for the inadvertent omission of the validation figures for the 138 reservoirs in our Res-CN dataset. We have taken corrective measures by uploading the figures to the link of our Res-CN data product (https://doi.org/10.5281/zenodo.7664489), and we kindly request that you access them from there.

Considering the extensive information contained within the supplementary file, we recognize the potential benefits of incorporating the validation figures - which, due to their size, span multiple pages - in our data product to facilitate user access and convenience. However, we also recognize the importance of maintaining a balance between completeness and conciseness in the main text. Consequently, we have presented only a subset of validations for select reservoirs alongside the overall evaluation accuracy. Nevertheless, we would like to assure users that all validation information is available in the data products. We are confident that this balance between completeness and conciseness is in line with the expectations of our readers, and we encourage them to refer to the data products for more detailed information.

The validation figures for all 138 reservoirs can be found in the "validation_figures" folder, which includes the time series of reservoir water level, water area, storage variation, and evaporation. In the "water level" directory, the time series of reservoir water level are available in two modes, i.e., high rate product and standard rate, along with their comprehensive evaluation reports and figures in PDF and TXT files. The "water area" directory provides the monthly area time series of reservoirs, accompanied by their comprehensive evaluation Excel files, including CC values compared with satellite-based water level, in situ water level, and other areal time series from other studies. Finally, the "storage variation" directory includes the time series and comprehensive evaluation figures in PDF files, which include statistical metrics.

Thank you for your feedback, and we hope that the inclusion of these validation figures will facilitate the use of our Res-CN dataset.

2. There are a lot of information provided by Res-CN, but some are not clearly explained. It mentioned in the introduction that 3,254 reservoirs were presented in this dataset, but in Table 2, the topography are available for 3,689 reservoirs. Table S10

summarized 18 attributes of topography, but it listed 19 attributes in Table 2. I can find 23 attributes in Table S13 for land cover. In addition, please clarify how the 173 is estimated from Table S14-S15 for the Soil & Geology. And how the 288 attributes are identified from Table S16 for Anthropogenic activity? Please clarify Table 2 and clearly link to the supplementary materials.

R1C2: Thank you for bringing up your concerns regarding the Table 2 and tables in the supplementary.

1. For attributes of topography, we indeed provided 19 attributes and made corrections in Table S10, and associated texts in the main text section 3.4.1.

Please find the modified Tables below.

**Table S10. Attributes of topography provided in the Res-CN.**

| Attribute | Unit | Description | Data source and reference |
|---|---|---|---|
| **length** | m | The length of the main stream measured from the basin outlet to the remotest point on the basin boundary. The main stream is identified by starting from the basin outlet and moving up the catchment. | Subramanya (2013) |
| **area** | km² | Calculated catchment area | Merri-Hydro (Yamazaki et al., 2019), GeoDAR (Wang et al., 2022) |
| **elev** | m | Mean catchment elevation | Merit-DEM (Yamazaki et al., 2019) |
| **elev_max** | m | Maximum catchment elevation | See above |
| **elev_min** | m | Minimum catchment elevation | See above |
| **elev_std** | m | Standard deviation of elevation in catchment | See above |
| **elev_range** | m | Range of catchment elevation (maximum minus minimum elevation) | See above |
| **slope** | m km⁻¹ | Mean catchment slope, Horn (1981) | See above |
| **mvert_dist** | km | Horizontal distance from the farthest point of the catchment to the corresponding gauge (length axis) | Merri-Hydro (Yamazaki et al., 2019) |
| **mvert_ang** | degree | Angle between the north direction and connection from farthest point of catchment to the corresponding gauge (length axis); e.g., direction from north (farthest catchment point) to south (gauge):180 degree, direction from east to west: 270 degree | See above |
| **elongation_ratio** | - | Ratio: elongation ratio, i.e., ratio between the diameter of an equivalent circle and the area of the catchment area to its length, Schumm (1956) | Subramanya, K. (2013) |
| **strm_dens** | km km⁻² | Ratio: stream density, i.e., ratio of lengths of streams and the catchment area | See above |

| resArea | km$^2$ | reservoir area. | Wang et al. (2022) |
| --- | --- | --- | --- |
| form_factor | - | Ratio: catchment area / (length)$^2$ | Subramanya, K. (2013) |
| shape_factor | - | Ratio: (catchment length)$^2$ / catchment area | See above |
| circulatory_ratio | - | Ratio: perimeter of the catchment / perimeter of the circle whose area is that of the basin | See above |
| compactness_coefficient | - | Ratio: perimeter of the catchment / perimeter of the circle whose area is that of the basin | See above |
| resArearatio | - | Ratio: reservoir area / catchment area | Merri-Hydro (Yamazaki et al., 2019), GeoDAR (Wang et al., 2022) |
| relief | - | Ratio: mean catchment elevation / Maximum catchment elevation | See above |

Main text: 19 topographic attributes are provided in Res-CN (Table S10).

2. For attributes of land cover, we indeed provided 23 attributes as shown in Table S13, we have corrected it to 23 in Table 2.
3. A total of 173 attributes pertaining to soil and geology are provided. Specifically, Table S14 presents 28 distinct soil attributes while Table S1 describes 19 geology attributes. Within the 28 soil attributes, 21 are represented across 7 levels encompassing six soil layers as well as the entire soil layer. An instance of this is the cation exchange capacity (CEC) of the soil, which has 7 associated attributes denoted as cec_1, cec_2, …, cec_6, and cec, indicating the CEC of the first to sixth soil layers and the entire soil layer. More explanations are added in the supplementary tables.

Please find the modified Tables below.

Table S14. Attributes of soil provided in the Res-CN.

| Attribute | Unit | Description | Data source |
| --- | --- | --- | --- |
| bdod* | kg dm-3 | Bulk density of the fine earth fraction | SoilGrids250 m (Hengl et al., 2017)[a] |
| cec* | cmol kg$^{-1}$ | Cation exchange capacity of the soil | See above |
| soc* | g kg$^{-1}$ | Soil organic carbon content in the fine earth fraction | See above |
| phh2o* | 10 | Soil pH | See above |
| pdep | cm | Soil profile depth | Shangguan et al. (2013) |
| cl | % | Percentage of clay content of the soil material | See above |
| sa | % | Percentage of sand content of the soil material | See above |
| por | cm$^3$ cm$^{-3}$ | Porosity | See above |

| | | | |
|---|---|---|---|
| **si** | % | Percentage of silt content of the soil material | See above |
| **grav** | % | Rock fragment content | See above |
| **som** | % | Soil organic carbon content | See above |
| **log_k_s*** | cm d$^{-1}$ | Log-10 transformation of saturated hydraulic conductivity | Soil hydraulic and thermal parameters (Dai et al., 2019)[a] |
| **theta_s*** | cm$^3$ cm$^{-3}$ | Saturated water content | See above |
| **tksatu*** | W m$^{-1}$ K$^{-1}$ | Thermal conductivity of unfrozen saturated soils | See above |
| **csol*** | J/(m$^3$K) | Volumetric heat capacity of soil solids in a unit soil volume | See above |
| **lambda*** | - | Pore size distribution index for the Campbell model | See above |
| **log_vgm_n*** | - | Log-10 transformation of a shape parameter for the VG model | See above |
| **psi_s*** | cm | Saturated suction for the Campbell model | See above |
| **tkdry*** | W m$^{-1}$ K$^{-1}$ | Thermal conductivity of dry soils | See above |
| **tksatf*** | W m$^{-1}$ K$^{-1}$ | Thermal conductivity of frozen saturated soils | See above |
| **vf_clay_s*** | cm$^3$ cm$^{-3}$ | Volumetric fration of clay | See above |
| **vf_gravels_s*** | cm$^3$ cm$^{-3}$ | Volumetric fration of gravel | See above |
| **vf_om_s*** | cm$^3$ cm$^{-3}$ | Volumetric fration of SOM | See above |
| **vf_quartz_mineral_s*** | cm$^3$ cm$^{-3}$ | Volumetric fration of quartz within mineral soils | See above |
| **vf_sand_s*** | cm$^3$ cm$^{-3}$ | Volumetric fration of sand | See above |
| **vf_silt_s*** | cm$^3$ cm$^{-3}$ | Volumetric fration of silt | See above |
| **vgm_alpha*** | cm$^{-1}$ | The inverse of the air-entry value for the VG model | See above |
| **vgm_theta_r*** | cm$^3$ cm$^{-3}$ | Residual moisture content for the VG model | See above |

* Within the aforementioned 28 soil variables, 21 variables marked with * are represented across 7 levels encompassing six soil layers as well as the entire soil layer. An instance of this is the cation exchange capacity (CEC) of the soil, which has 7 associated attributes denoted as cec_1, cec_2, …, cec_6, and cec, indicating the CEC of the first to sixth soil layers and the entire soil layer, i.e., at six layers of 0–0.05, 0.05–0.15, 0.15–0.30, 0.30–0.60, 0.60–1.00, and 1.00–2.00m, as well as the whole soil layer. In this sense, we provided 154 soil attributes.

Main text: Res-CN provided 154 attributes to characterize physical and chemical properties of soil (Tables S14).

4. A total of 288 attributes pertaining to soil and geology are provided. Within the population category, there are five included attributes, namely population_2000, population_2005, population_2010, population_2015, and population_2020. As for the Nighttime light category, which comprises of "avg_vis", "stable_lights", "cf_cvg", and "avg_lights_x_pct", both the mean and

sum values for each variable are provided for all available time frames. To illustrate, the variable mean_cf_cvg_101994 denotes the mean value of cf_cvg for the month of October in 1994. Accordingly, a total of 288 anthropogenic attributes have been provided. More explanations are added in the supplementary tables.

Please find the modified Tables below.

**Table S16. Attributes of anthropogenic activity provided in the Res-CN.**

| Attribute | Unit | Description | Data source |
|---|---|---|---|
| **population*** | - | Population for the years 2000, 2005, 2010, 2015, and 2020 | Gridded Population of the World (GPW) database v4.11 |
| **avg_vis*** | - | The average of the visible band digital number values with no further filtering | DMSP-OLS Nighttime Lights v4 dataset (Doll, 2008) |
| **stable_lights*** | - | The cleaned up avg_vis contains the lights from cities, towns, and other sites with persistent lighting, including gas flares. Ephemeral events, such as fires, have been discarded. The background noise was identified and replaced with values of zero | See above |
| **cf_cvg*** | - | Cloud-free coverages tally the total number of observations that went into each 30-arc second grid cell. This band can be used to identify areas with low numbers of observations where the quality is reduced. | See above |
| **avg_lights_x_pct*** | - | The average visible band digital number (DN) of cloud-free light detections multiplied by the percent frequency of light detection. The inclusion of the percent frequency of detection term normalizes the resulting digital values for variations in the persistence of lighting. For instance, the value for a light only detected half the time is discounted by 50%. Note that this product contains detections from fires and a variable amount of background noise | See above |
| **reproject_grip4_total_dens_m_km2** | m km$^{-2}$ | Road density | Global Roads Inventory Project (GRIP) dataset (Meijer et al., 2018) |
| **reproject_hfp2009** | - | The Human Footprint camp of cumulative pressures on the environment in 2009 | Global Human Footprint v2 dataset (Venter et al., 2016) |
| **reproject_hfp1993** | | The Human Footprint camp of cumulative pressures on the environment in 1993 | Global Human Footprint v2 dataset (Venter et al., 2016) |

* Within the population category, there are five included attributes, namely population_2000, population_2005, population_2010, population_2015, and population_2020. As for the Nighttime light category, which comprises of avg_vis, stable_lights, cf_cvg, and avg_lights_x_pct, both the mean and sum values for each variable are provided for all available time frames. To illustrate, the variable mean_cf_cvg_101994 denotes the mean value of cf_cvg for the month of October in 1994. Accordingly, a total of 288 anthropogenic attributes have been provided.

5. Yes, we indeed provide all data for 3254 reservoirs. In this study, we delineated reservoir upstream catchment and provided two types of catchments, i.e., full

catchment and intermediate catchment. Res-CN provides 3254 full catchments and 435 intermediate catchments (See Fig. 2). So, that's why for catchment-level attributes the number should be 3254 full catchments + 435 intermediate catchments.

**a) Basin delineation A | Full catchments**   **b) Basin delineation. B | Intermediate catchments**

[Figure]

**Figure 2.** Types of catchment delineation in Res-CN shown with an example. (a) Catchment delineation A: full catchments, which are defined as the full upstream contributing area of a reservoir. In plot (a), the area of reservoir 23720 overlaps with that of reservoir 3205 and that of 6651. (b) Catchment delineation B: intermediate catchment. In plot (b), all upstream contributing areas of the upstream reservoirs (3205 and 6651) are removed from the full catchment of reservoir 23720, thus, we get the intermediate catchment of reservoir 23720 (in black boundary). Background in light blue indicates other catchments not shown in this example. Source of background: MERIT Hydro and MERIT DEM (Yamazaki et al., 2019).

In summary, we have carefully checked and made corrections in the Table 2 and all tables in supplementary. Please find the modified Table 2 as well.

**Table 2. Summary of the data provided in the Res-CN.**

|  | **Variable** | **Number of (reservoirs/catchments)** | **Description** |
|---|---|---|---|
| **Time series of reservoir states** | Water level (SR, a total of 650 reservoirs) | 54 | From Jason-3 mission, 10-days, 2016-2022, with 3 retracking algorithms |
|  |  | 192 | From Sentinel-3A mission, 27-days, 2016-2022, with 5 retracking algorithms |
|  |  | 194 | From Sentinel-3B mission, 27-days, 2018-2022, with 5 retracking algorithms |
|  |  | 215 | From ICESat-2 mission, 90-days, 2019-2022, with 1 retracking algorithm |
|  |  | 347 | From CryoSat-2 mission, 369-days, 2010-2022, with 3 retracking algorithms |

| | | 229 | From SARAL/AltiKa mission, 35-days, 2016-2022, with 5 retracking algorithms |
|---|---|---|---|
| | Water level (HR) | 250 | High rate (HR) product by merging SR products, from 2010-2020, sub-monthly or monthly |
| | Water area | 3214 | Monthly from 1984-2021 |
| | Storage variation | 2999 | Monthly storage variation from 1984-2021 |
| | Evaporation | 3185 | Monthly evaporation rate and volume from 1984-2021 |
| **Catchment-level attributes** | Reservoir and catchment shapefile | 3254 full catchments, 435 intermediate catchments | Two types of reservoir upstream catchments, catchment shapefile attributes (Tables S9-10) |
| | Topography | 3254 full catchments, 435 intermediate catchments | 19 attributes (Table S10) |
| | Climate data | Same as above | 11 climatic attributes and daily time series of metrological data with 15 variables from 1980-2022 (Tables S11-12) |
| | Land cover | Same as above | 23 attributes (Table S13) |
| | Soil & Geology | Same as above | 173 attributes (Tables S14-15) |
| | Anthropogenic activity | Same as above | 288 attributes (Table S16) |

3. It will be useful to add more details for the water areas at line 88. For example, the range of 0.004-1373.77 [km²] is very wide. A histogram of the water areas will be useful for the end-users because researchers have different focuses. For example, a watershed hydrologist may be interested in relatively small reservoirs, but an Earth system modeler may only need large reservoirs. Also, it will be helpful to list some major reservoirs based on the water areas (e.g., first ten or twenty?). As argued by the author, the largest reservoir area is 1,373.77 [km²], but this number is not consistent with my source.

R1C3: We have added the histogram of the water areas and listed out top 10 reservoirs based on the water areas in the supplementary file. In the data product, apart from the shapefile, we added one more excel file to list all reservoirs attributes such as reservoir's area, Chinese name, and GeoDAR ID.

See added figure below:

[Figure]

**Figure S7.** Distribution of reservoir area values and Top 20 reservoirs based on area size in our data product. For more information such as area, name and ID of all reservoirs, please refer to our data product.

In this study, we focused on reservoirs for which are mapped and available from the newest global GeoDAR database (Wang et al., 2022, https://doi.org/10.5281/zenodo.6163413). We checked the GeoDAR again, and found

that the largest reservoir area in China is 1,373.77 [$km^2$]. To clarify this issue, we changed the text as follows:

In this study, we constructed reservoir-catchment characteristics for 3254 reservoirs recorded in the GeoDAR database (Wang et al., 2022), with water areas ranging from 0.004 and 1373.77 $km^2$(Fig. S7), with a total water storage capacity of 682,595 $km^3$ accounting for 73.2% Chinese reservoir water storage capacity.

References:

Wang, J., Walter, B. A., Yao, F., Song, C., Ding, M., Maroof, A. S., Zhu, J., Fan, C., McAlister, J. M., Sikder, S., Sheng, Y., Allen, G. H., Crétaux, J.-F., and Wada, Y.: GeoDAR: georeferenced global dams and reservoirs dataset for bridging attributes and geolocations, Earth Syst. Sci. Data, 14, 1869–1899, https://doi.org/10.5194/essd-14-1869-2022, 2022.

**Minor Comments:**

Line 42, Please capitalize **E**arth.

Changed as suggested.

Line 76-78: "In addition, there is no systematic assessment of whether reservoir water levels or water areas from previous studies and databases agree with one another, as shown in this study by many reservoirs whose in situ measurements are available.". I don't understand this statement, are you trying to argue your results suggests the results from previous studies are biased when compared to in-situ measurements?

Yes, rather than biases, to be fair, we just mentioned that there are some differences among these datasets. Thus, we say try our best to do some cross comparison and validation against gauged measurements.

For example, for our area dataset that using a algorithm developed by Donchyts et al. (2022), we compared them with water level time series (in situ and altimetric measurements) and the water level of two other, similar areal datasets: i.e., GRSAD (Zhao and Gao, 2018) and ReaLSAT (Khandelwal et al., 2022). Based on all the compared reservoirs available, we found that our SWA time series show good agreement to values in GRSAD (median CC value of 0.64, rBIAS=−9%, rRMSE=26%, and n=338) and ReaLSAT (median CC value=0.68, rBIAS=−10%, rRMSE=22%, and n=47) datasets. See figure Fig. S3 below: Overall, these comparisons suggest a good level of trustworthiness in our water area time series.

[Figure]

**Figure S3. Graphs showing reservoir water area time series against in situ water levels, altimetric water levels from high and standard rates, and GRSAD and ReaLSAT area time series for a sample of reservoirs of varying areas.**

For water level, we validation against in situ data and three similar datasets, finding some differences, see figure Fig. S2 below. We argue that some differences can be found when comparing them together. some large discrepancies can be found in certain reservoirs, e.g., the Shuifeng reservoir (Fig. S2. 16) did not show a clear fluctuation pattern as captured by G-REALM, for the periods in 2020 between our dataset and Hydroweb at the Fengman reservoir (Fig. S2. 3). Our datasets are denser than Hydroweb over most reservoirs (Fig. S2. 5) and can be less noisy. These advantages would benefit the continuity and accuracy of the remotely sensed WSE and RWSC. Overall, this comparison demonstrated that performance of our datasets approximates accuracy of existing global altimetry datasets.

[Figure]

**Figure S2. Comparison between our water level time series and other existing similar databases.**

References:

Donchyts, G., Winsemius, H., Baart, F., Dahm, R., Schellekens, J., Gorelick, N., Iceland, C., and Schmeier, S.: High-resolution surface water dynamics in Earth's small and medium-sized reservoirs, Sci. Rep., 12, 13776, https://doi.org/10.1038/s41598-022-17074-6, 2022.

Khandelwal, A., Karpatne, A., Ravirathinam, P. Ghosh, R., Wei. Z., Dugan, H. A., Hanson, P. C., and Kumar, V.: ReaLSAT, a global dataset of reservoir and lake surface area variations, Sci. Data, 9, 356, https://doi.org/10.1038/s41597-022-01449-5, 2022

Zhao, G. and Gao, H.: Automatic Correction of Contaminated Images for Assessment of Reservoir Surface Area Dynamics, Geophys. Res. Letters, 45, 6092–6099, https://doi.org/10.1029/2018GL078343, 2018.

Line 80: "there are approximately 30 Chinese" Do you mean there are approximately 30 reservoirs from China?

Changed as: there are approximately 30 Chinese reservoirs.

Line 106: Please provide the source or reference for the number of 98,000.

Yes, we added the reference below.

References:

MWR: Hydrologic Data Yearbook, Ministry of Water Resources (MWR), ISBN 9771009737167, 2016.

Line 109: Are the 3,254 reservoirs from GeoDAR?

Yes, we used the reservoirs shapefiles from GeoDAR.

Line 135: What is your criteria for reservoirs with large variations.

The threshold used in our study was obtained based on previous research (Jiang et al. 2017 RSE). However, as our study covers many reservoirs, some of which may experience water level fluctuations exceeding 40 meters, we adjusted the threshold for certain reservoirs. In fact, we set a series of thresholds, such as 20, 30, 40, and 50 m, for each reservoir. We found that this parameter was not sensitive because the method used in the next step estimates along-track water level in the presence of outlying measurements (Nielsen et al. 2015).

References:

Liguang Jiang, Karina Nielsen, Ole Baltazar Andersen, Peter Bauer-Gottwein, CryoSat-2 radar altimetry for monitoring freshwater resources of China, Remote Sensing of Environment, 200, 2017, 125-139, https://doi.org/10.1016/j.rse.2017.08.015.

Nielsen, K., Stenseng, L., Andersen, O. B., Villadsen, H., and Knudsen, P.: Validation of CryoSat-2 SAR mode based lake levels, Remote Sens. Environ., 171, 162–170, https://doi.org/10.1016/j.rse.2015.10.023, 2015.

Line 165: I am confused about this statement. Is this "768 reservoirs" from this study? If so, please clarify it. If not, please cite reference to support it.

No, these reservoirs are from Spain, India, South Africa, and the USA, and the algorithm is validated using data from 768 reservoirs located in these four countries. We have revised the sentence as follows:

The algorithm has been applied to map water areas in 768 reservoirs of different sizes and climate zones located in Spain, India, South Africa, and the USA, and there is strong evidence to suggest that it performs well in this regard (Donchyts et al., 2022).

References:

Donchyts, G., Winsemius, H., Baart, F., Dahm, R., Schellekens, J., Gorelick, N., Iceland, C., and Schmeier, S.: High-resolution surface water dynamics in Earth's small and medium-sized reservoirs, Sci. Rep., 12, 13776, https://doi.org/10.1038/s41598-022-17074-6, 2022.

Line 243-244: Add reference or results to show the validation of delineation for the 1,398 catchments.

Yes, we added the reference below.

References:

Xie, J., Liu, X., Bai, P., and Liu, C.: Rapid watershed delineation using an automatic outlet relocation algorithm, Water Resour. Res., 58, e2021WR031129, https://doi.org/10.1029/2021WR031129, 2022.

Line 305-306: The authors explain the large errors occurs in 55 catchments are because the size of the catchments is small. But Figure 3d and f show the large errors also occur in large reservoirs. The spatial map is not very clear to show where the errors from. Consider plotting the comparison with the reference dataset using the scatter plot.

A scatter plot is added in our supplementary file. The explanations can be found below.

**Main text:** To compare Res-CN with GRanD and LakeATLAS, we spatially joined reservoir shapefiles from both datasets, matching reservoirs that overlapped for greater than 90% of their extent. Based on this subset of reservoirs, we found that catchment areas delineated in this study corresponded relatively well to catchment areas in both GRanD (CC = 0.999, n = 910) and LakeATLAS (CC = 0.910, n = 2147), which proves the reliability of our delineated catchments. Large discrepancies occur in 55 catchments, whose absolute relative error is greater than 100% (Fig. 3e, f). Small reservoirs located near confluences between rivers of different sizes are more likely to be affected by this issue, as a minor spatial mismatch can assign a reservoir to the small catchment of the tributary stream rather than the large catchment of the mainstream, and vice versa (Fig. S8). The differences in catchment delineation between these datasets result from differences in both DEM and methods for flow direction correction and depression filling and pour points correction. In this study, the widely verified MERIT Hydro flow directions are used, and we suggest that cautions should be taken when using catchments with large error discrepancies with LakeATLAS, which is based on the drainage direction grids of HydroSHED (Fig. S8a).

[Figure]

Figure S8. Comparison of the areas of delineated catchments in this study with those of LakeATLAS (Lehner et al., 2022), and those of GRanD reported value (Lehner et al., 2011).

References:

Lehner, B., Messager, M. L., Korver, M. C. and Linke, S.: Global hydro-environmental lake characteristics at high spatial resolution. Sci. Data, 9, 351, https://doi.org/10.1038/s41597-022-01425-z, 2022.

Lehner, B., Liermann, C. R., Revenga, C., Vörösmarty, C., Fekete, B., Crouzet, P., Döll, P., Endejan, M., Frenken, K., Magome, J., Nilsson, C., Robertson, J. C., Rödel, R., Sindorf, N., and Wisseret, D.: High-resolution mapping of the world's reservoirs and dams for sustainable river-flow management, Front. Ecol. Environ., 9, 494–502, https://doi.org/10.1890/100125, 2011.

Line 326-328: I am not sure if RMSE is a good metric to indicate error for the water level. The magnitude of water level varies with reservoir size. So, RMSE = 0.3m is considered as small error for a large reservoir, but it can be significant for a small reservoir. Since the time series of water levels are compared, some evaluation metric like NSE or KGE can provide more information about the evaluation.

I partially agree with your suggestions. The root mean square error (RMSE) is a common practice in satellite altimetry research, as evidenced by several references listed below. It is important to note that satellite altimetry measurements have a relatively coarse resolution, usually on a monthly or sub-monthly basis, which is why other metrics such as the Nash-Sutcliffe efficiency (NSE) or Kling-Gupta efficiency (KGE) are seldom used in this field. Nonetheless, we provide users with both the correlation coefficient (CC) value and time series of PDF figures for each reservoir to consider.

References:

Gao, H., Birkett, C., and Lettenmaier, D. P.: Global monitoring of large reservoir storage from satellite remote sensing, Water Resour. Res., 48, W09504, https://doi.org/10.1029/2012WR012063, 2012.

Jiang, L., Nielsen, K., Dinardo, S., Andersen, O. B., and Bauer-Gottwein, P.: Evaluation of Sentinel-3 SRAL SAR altimetry over Chinese rivers, Remote Sens. Environ., 237, 111546, https://doi.org/10.1016/j.rse.2019.111546, 2020.

Tourian, M. J., Elmi, O., Shafaghi, Y., Behnia, S., Saemian, P., Schlesinger, R., and Sneeuw, N.: HydroSat: geometric quantities of the global water cycle from geodetic satellites, Earth Syst. Sci. Data, 14, 2463–2486, https://doi.org/10.5194/essd-14-2463-2022, 2022.

Vu, D. T., Dang, T. D., Galelli, S., and Hossain, F.: Satellite observations reveal 13 years of reservoir filling strategies, operating rules, and hydrological alterations in the Upper Mekong River basin, Hydrol. Earth Syst. Sci., 26, 2345–2364, https://doi.org/10.5194/hess-26-2345-2022, 2022.

Line 330: There is no Fig. S7 in supplementary materials.

Sorry for this. It should be Fig. S1.

Line 335: There is no Fig. S8 in supplementary materials.

Sorry for this. It should be Fig. S2.

Line 372-373: Fig.6a and b plot the water areas comparisons from all the reservoirs and months, then what does the median CC mean?  Did you also estimate the CC for each reservoir? Please clarify what does the median CC mean. Also, it is critical to show the evaluation at site level to demonstrate the accuracy of Res-CN.

Hope the R1C1 reponse addressed your concern. The validation figures for all 138 reservoirs can be found in the "validation_figures" folder. We hope that the inclusion of these validation figures will facilitate the use of our Res-CN dataset.

For each reservoir, we calculate the correlation coefficient (CC) value and determine the median value of all the CC values. Therefore, the median CC refers to the median of these individual CC values.

Line 384: NRMSE, CC and RMSE have median values of 21%,0.53, and 0.03 $km^3$, respectively.

Changed as suggested. Thank you very much.

Line 391: Please specify the number of available reservoirs when using the water areas and water levels to derive the storage variations. Are they the same reservoirs that used the DEM's area-storage model?

We have added this information in the sentence:

To solve this problem, we provide another type of storage variation estimates for 335 reservoirs using satellite water areas and water levels (see section 2.3, Shen et al., 2022b).

Yes, they are the same reservoirs as all reservoirs have the storage variation estimates that used the DEM's area-storage model.

Line 412: Please clarify this sentence:" Long-term mean meteorological variables calculated the evaporation rates are available in Fig.S9."

The figure should be Fig. S3. We have added this information in the sentence:

Long-term mean meteorological variables that were used to calculate the evaporation rates are depicted in Fig. S3.

Line 424: Consider changing the colormap for Figure 8b, because the map doesn't show any variation of water areas (e.g., only blue shows up).

We have replotted this figure as follows.

[Figure]

**Figure 8.** Validation of reconstructed monthly reservoir evaporation values. (a and b) Long-term mean evaporation rates and water areas during 1984-2020. (c) Comparison between constructed monthly reservoir evaporation and observed pan evaporation values at the Danjiangkou reservoir. (d) Seasonal cycle.

Line 533: Were machine learning methods used in this study to derive the soil properties at different depths? If so, please specify what algorithm was used and how it

was applied in this study. If machine learning methods were used in existing dataset to derive the soil properties, please clarify it.

Yes, we just used the existing dataset that are based on the machining learning methods. Sorry for the confusion and we have clarified the sentences as follows.

The SoilGrids250 dataset predicted soil properties at six different soil layers (i.e., 0-0.05m, 0.05-0.15m, 0.15-0.3m, 0.3-0.6m, 0.6-1m, and 1-2m) using machine learning techniques, utilizing data from approximately 150,000 soil profiles and 158 environmental covariates derived from remote sensing data on a global scale.

Line 593: "Earth".

Changed as suggested.

---

## Author Comment (AC2)

**Reviewer #2 Comment on essd-2022-422 (Anonymous Referee #2)**

Dear Anonymous Referee #2,

Thank you for your time and efforts in reviewing our manuscript. Please find attached point-to-point responses regarding your comments (marked in purple) and made corresponding changes in the main manuscript (in red). We hope that the improved manuscript can help the readers to better understand our study.

Kind regards.

**Summary:**

This study presents time series data on hydrometeorological, topographic, and catchment attributes for over 3000 Chinese reservoirs. The authors have brought together datasets from many disparate sources, including in-situ data/information and satellite products, which is a commendable effort. The methods used are technically sound and the final product derived could be of great value for many purposes including hydrological modeling, water resource management, and ecosystem studies. The results presented provide many insights on reservoir attributes with a large spatial and temporal coverage. Therefore, this study is worthy of publication; however, there are certain issues that require further attention. In terms of presentation quality, the paper is generally well written but is not devoid of certain typos, grammatical errors, unclear statements. The authors should very carefully proofread the entire manuscript before submitting it again. My overall assessment is that the paper can be published after major revisions. I provided my detailed comments below.

R2C0: Thank you for your recognition of the strengths of our study. We appreciate your constructive feedback and have carefully considered all of your suggestions and comments. We agree that the paper required attention to certain issues, such as typographical errors, grammatical mistakes, and unclear statements. We have thoroughly proofread the manuscript and made the necessary revisions to address all of your concerns.

We hope that these revisions have improved the overall quality of the manuscript. We are grateful for your time and expertise in reviewing our work, and we believe that your feedback has made a valuable contribution to the study's scientific value. Thank you again for your comments, and we hope that the revised manuscript will meet your expectations.

**Major comments:**

L66: I suggest rephrasing the statement, especially for "failed". The many studies noted by the authors have substantially advanced our ability to better monitor and model reservoirs globally. Perhaps, the datasets could be incomplete and there are more opportunities to develop relatively more comprehensive datasets, but I suggest giving a bit more positive bend to this statement; "failed" seems a bit unfair!

R2C1: We agree that the previous efforts mentioned in our manuscript have substantially advanced our understanding of global reservoir monitoring and modeling. Our aim was to highlight the potential for relatively more comprehensive datasets. Upon reflection, we acknowledge that the term "failed" may be overly negative and unfair. We have rephrased this statement in the revised manuscript to reflect our intent more accurately and to give a more positive bend. We have now emphasized the opportunities for further improvements in data collection and highlighted the potential for even more extensive datasets in the future. Thank you for your feedback, and we appreciate the opportunity to improve the clarity and accuracy of our manuscript. Upon further consideration of the context, we have decided to delete the statement as it was deemed inappropriate.

L89-90: some modeling studies that have dealt with such challenges could be cited here including (Dang et al., 2022; Dang et al., 2020; Galelli et al., 2022; Shin et al., 2020)

R2C2: Yes, thanks for your kind reminder and we carefully checked that all studies mentioned are highly related to our reservoir datasets. We cited all these studies in the section of Introduction, Summary and application.

See main text below:

Results of this study facilitated managements of reservoirs and relevant studies such as hydrological modeling, environmental studies, and climate research in the spatially explicit context of reservoir catchment-level (Dang et al., 2020; Galelli et al., 2022).

This is particularly true if the reservoir inflow is also utilized. Recently, the gridded natural runoff provided by Gou et al. (2021) provides exciting opportunities for quantifying the human water regulation in combination with Res-CN (Dang et al., 2022; Shin et al., 2020).

**References**

Dang, H., Pokhrel, Y., Shin, S., Stelly, J., Ahlquist, D., Du Bui, D., 2022. Hydrologic balance and inundation dynamics of Southeast Asia's largest inland lake altered by hydropower dams in the Mekong River basin. Science of The Total Environment, 831: 154833.

Dang, T.D., Vu, D.T., Chowdhury, A.K., Galelli, S., 2020. A software package for the representation and optimization of water reservoir operations in the VIC hydrologic model. Environmental Modelling & Software, 126: 104673.

Galelli, S., Dang, T.D., Ng, J.Y., Chowdhury, A., Arias, M.E., 2022. Opportunities to curb hydrological alterations via dam re-operation in the Mekong. Nature Sustainability: 1-12.

Shin, S., Pokhrel, Y., Yamazaki, D., Huang, X., Torbick, N., Qi, J., Pattanakiat, S., Ngo-Duc, T., Nguyen, T.D., 2020. High Resolution Modeling of River-Floodplain-Reservoir Inundation Dynamics in the Mekong River Basin. Water Resources Research, 56(5): e2019WR026449.

L84: what does "states" mean here?

R2C3: we have reprahsed as: In addition to the time series of reservoir datasets described above,

L85 and elsewhere: I don't think a "catchment shapefile" is a "catchment attribute"; file is a file. There are many other such instances where certain terminologies are not properly used. Also, what the "anthropogenic activity" – used in a singular form implies there is one such activity that is being considered.

R2C4: Thank you for your comments regarding the terminology used in our manuscript. We appreciate your keen attention to detail and agree that the terminology used should be precise and accurate. Upon review, we agree that the term "catchment attribute" was not an appropriate descriptor for the catchment shapefile used in our study, and we have removed "catchment shapefile" here and revised the main text accordingly.Regarding the use of the term "anthropogenic activity", we apologize for any confusion this may have caused. We have revised as "anthropogenic activity characteristics" in the manuscript to better reflect the multiple anthropogenic activities that were considered in our analysis.

We appreciate your feedback and attention to detail, and we believe that the revisions we have made will improve the clarity and accuracy of our manuscript.

Section 2.1: Why was 10% threshold used for the GSW data? The same question applies for 20 and 40 meters. Please provide justification. Further, I could imagine all of the many products used in these methods contain substantial uncertainties (being primarily remote sensing based). How would those uncertainties affect the outcomes derived here and how did the authors deal with these issues?

R2C5: A low threshold of 10% is chosen for two reasons: (1) Water occurrences are expected to be low for the newly built reservoirs and (2) a lower threshold ensures that a higher number of potential measurements are preselected. We provided a reference here for justification (Zhang et al., 2020).
The threshold of 20 and 40 meters was set in previous studies (Jiang et al. 2017 RSE). In fact, we set a series of thresholds, such as 20, 30, 40, and 50 m, for each reservoir. Interestingly, we found that this parameter was not sensitive because the method of tsHydro (https://github.com/cavios/tshydro) used in the next step estimates alongtrack water level in the presence of outlying measurements (Nielsen et al. 2015), and also provides the uncertainties for each value in the time series. You can be found in the corresponding data product file. For example, in the "D reservoir states"/"water level"/Standard Rate/OBS/S3A/, the csv files contains, "year", "month", "demical_year","s3a_wl", and "s3a_wlsd".

We have discussed the sources of uncertainties, and their impacts in each section (considered your comments below).

Main text:

- Section 3.3.1: We provided the uncertainty information for each value of the time series in the data product file. The SD (standard deviation) estimates can quantify the accuracy of the water level along the track at the level of individual data points (Fig. S8). Water level time series for each reservoir are available in Rec-CN as EXCELs, PDFs and detailed evaluation reports based on in situ data when available (see Section of data availability).

[Figure]

**Fig. S8.** Uncertainties for each value in the time series of reservoir water level. In the figure, black line refers to the observed water level, black dot refers to altimetric water level, error bar quantifies the uncertainty of each value. Taking 20 reservoirs in the Standard rate product as an example, 1-4 are taken from Jason-3 mission, 5-8 are from SARAL/AltiKa mission, 9-12 are from Sentinel-3A mission, 13-16 are from Sentinel-3B mission, 17-20 are from CryoSat-2 mission. All uncertainties values are available in our product.

- Section 3.3.2: Uncertainties in surface water area estimates are generally attributed to satellite images and algorithms. As reported by Zhao et al. (2022), the uncertainty of Landsat-based GRSAD areal dataset is 6.1%. In this study, we generated a more reliable reservoir water area product by fusing both Landsat and Sentinel-2 images (Fig. S9), using an algorithm that can largely reducing the impacts of cloud contaminations (Donchyts et al., 2022). There is strong evidence to suggest that this algorithm performs well in this regard, as it has been widely validated in 768 reservoirs of different

sizes and climate zones located in Spain, India, South Africa, and the USA (Donchyts et al., 2022). Nevertheless, some limitations and future developments should be considered. Our first option is to use Sentinel-1 data to provide more information in cloudy regions. Furthermore, the algorithm may be improved by either multiclass Otsu or using advanced machine learning methods.

[Figure]

**Figure S9.** Graphs showing reservoir water area time series against in situ water levels, altimetric water levels from high and standard rates, and GRSAD and ReaLSAT area time series for a sample of reservoirs of varying areas (Shen et al., 2022b).

- Section 3.3.3: The uncertainties in storage anomalies are primarily attributed to three sources, i.e., the altimetric water level, water surface area estimations from Landsat and Sentinel-2 images, and the error resulting from their combination (the hypsometric curve). Fig. S10 provides an example that illustrates how the uncertainties in satellite datasets propagate to storage anomalies. According to Shen et al. (2022), the primary source of error in storage anomaly is water surface area and the hypsometric curve. Regarding the water surface area, after applying the algorithm developed by Donchyts et al. (2022), these errors and impacts can be reduced to a large extent. Meanwhile, we employed five hypsometric relationships, and the one with the highest $R^2$ value for further use. For more than 80 % reservoirs, the $R^2$ values are greater than 0.5, providing a strong foundation for storage anomaly estimates. Nonetheless, the current satellite sensors have limitations, as evidenced by the significant discrepancies observed in peak values (Figure 7). The increasing temporal resolution and data accuracy of satellite datasets, such as the SWOT mission, will likely improve the accuracy of storage anomaly estimates in the future.

[Figure]

**Figure S10.** Graphs showing an example that illustrates how the uncertainties in satellite datasets propagate to storage anomalies. Error series and relationships of reservoir elevation-storage. Error series of (a) SWE-derived RWSC (i.e., storage anomaly), (b) WSE-derived RWSC and water level change, (c) WSE (i.e., water level). (d) and (e) Relationships of elevation-storage. The numbers on the x-axis (a, b, c) refer to the IDs of SWE, WSE, and WSE change observations, respectively. For more details about the propagation process, please find the reference Shen et al., (2020): https://doi.org/10.3390/rs14040815.

- ## References:

Liguang Jiang, Karina Nielsen, Ole Baltazar Andersen, Peter Bauer-Gottwein, CryoSat-2 radar altimetry for monitoring freshwater resources of China, Remote Sensing of Environment, 200, 2017, 125-139, https://doi.org/10.1016/j.rse.2017.08.015.

Nielsen, K., Stenseng, L., Andersen, O. B., Villadsen, H., and Knudsen, P.: Validation of CryoSat-2 SAR mode based lake levels, Remote Sens. Environ., 171, 162–170, https://doi.org/10.1016/j.rse.2015.10.023, 2015.

Shen, Y., Liu, D., Jiang, L., Tøttrup, C., Druce, D., Yin, J., Nielsen, K., Bauer-Gottwein, P., Wang, J., and Zhao X.: Estimating reservoir release using multi-source satellite datasets and hydrological modeling techniques, Remote Sens., 14, 815, https://doi.org/10.3390/rs14040815, 2022.

Zhao, G., Li, Y., Zhou, L., and Gao, H.: Evaporative water loss of 1.42 million global lakes. Nat. Commun., 13, 3686, https://doi.org/10.1038/s41467-022-31125-6, 2022.

Donchyts, G., Winsemius, H., Baart, F., Dahm, R., Schellekens, J., Gorelick, N., Iceland, C., and Schmeier, S.: High-resolution surface water dynamics in Earth's small and medium-sized reservoirs, Sci. Rep., 12, 13776, https://doi.org/10.1038/s41598-022-17074-6, 2022.

The comment above regarding uncertainty applies to Sections 2.2 and 2.3 as well. I suggest that the authors discuss various uncertainty sources and their impacts.

R2C6: We appreciate the reviewers' insightful and helpful comments on our manuscript. We have revised the manuscript according to the reviewer's suggestion. We have discussed the uncertainties of the dataset in the revised manuscript (section 3.3.1-3.3.3) to facilitate the usage of this dataset. We did our best to collect the most reliable datasets to date and will regularly update the related datasets in the future to ensure their timeliness. Hope R2C5 response addressed your concern.

Figure 3 caption: please add unit to the x-axis of the histograms or provide a note in the caption. I wondered why the panels are organized in this specific order – why not swap (e) and (f) so that the same categories sit adjacent to each other.

R2C7: We have changed the figure 3 as suggested.

[Figure]

Fig. 3. Distribution of the delineated catchments (intermediate catchments and full catchments). Each category's histogram indicates the number of basins (out of 3254). In a histogram, the X-axis represents the number of basins, while the Y-axis represents each subplot's title. Circle sizes are proportional to catchment areas.

Figure 4 and others: The Zenodo link was not active, so I couldn't make sure if all the datasets were shared. Are all in-situ datasets included in the publicly shared database?

R2C8: From my location in Japan, I have verified the accessibility of the Zenodo link (https://doi.org/10.5281/zenodo.7664489). I apologize for any inconvenience caused. All data presented in the figures and tables has been shared on Zenodo, with the exception of certain in situ reservoir water level and storage data.
We obtained daily water level and storage data spanning 2015–May 2021 for 93 reservoirs from the local watershed agency (http://113.57.190.228:8001/web/Report/BigMSKReport, last access: 15 October 2022) and National Hydrological Information Centre for validation (http://xxfb.mwr.cn/index.html, last access: 15 October 2022).
However, the in-situ datasets are updated day-by-day, thus, not possible to download the historical time series. I apologize for not making our collected in-situ datasets publicly available on Zenodo as we have a federal grant that limits the sharing of in-situ dataset. Moreover, we have no right to make all of them publicly available, now. Anyway, we are happy to share most of these data for users to do some case studies, please feel free to contact the corresponding author (yjshen2020@gmail.com).

Figure 7: Why does Res-CN under or overshoot storage for many reservoirs (e.g., panels 7,8 etc.)?

R2C9: Yes, we add more explanations and discussed the uncertainties as well as limitations in this section. Please also note that for Fig. 7 panels 9-12, our data indeed captured the large peak values for most reservoirs (2, 0.5, 0.2 km$^3$).

Main text: The Res-CN database provides monthly reservoir water storage anomaly for 3254 Chinese reservoirs during 1984-2020 using DEM's area-storage model, along with their detailed evaluation reports (see Section of data availability).

The remotely sensed storage anomalies generally agree with the observations represented by the statistical metrics, although some large discrepancies occur in peak values.

The uncertainties in storage anomalies are primarily attributed to three sources, i.e., the altimetric water level, water surface area estimations from Landsat and Sentinel-2 images, and the error resulting from their combination (the hypsometric curve). Fig. S10 provides an example that illustrates how the uncertainties in satellite datasets propagate to storage anomalies. According to Shen et al. (2022), the primary source of error in storage anomaly is water surface area and the hypsometric curve. Regarding the water surface area, after applying the algorithm developed by Donchyts et al. (2022), these errors and impacts can be reduced to a large extent. Meanwhile, we employed five hypsometric relationships, and the one with the highest $R^2$ value for further use. For more than 80 % reservoirs, the $R^2$ values are greater than 0.5, providing a strong foundation for storage anomaly estimates. Nonetheless, the current satellite sensors have limitations, as evidenced by the significant discrepancies observed in peak values (Figure 7). The increasing temporal resolution and data accuracy of satellite datasets, such as the SWOT mission, will likely improve the accuracy of storage anomaly estimates in the future.

[Figure]

**Figure 7.** Time series of water surface area and storage anomaly in selected reservoirs. RMSE (km³), NRMSE, and CC values are given at the top of each subplot when in situ observations available. Note that: time series of water surface area and storage anomaly of the remaining reservoirs are available in our datasets.

Figure 8: I can't really tell whether this is a good/bad match between the three? I suggest adding some statistical measures such as RMSE and also a seasonal climatology panel on the right (could be just for the period with observed data).

R2C10: We adopted the validation from Tian et al., (2021) for evaluation of reservoir evaporation product considering we found that our pan evaporation is not the observed evaporation, and we cannot provide the source of this dataset. Thus, revised the figure 8 and re-create figure s11 for validation. Please find our revised text below:

Res-CN provides monthly reservoir evaporation values for 3254 Chinese reservoirs during 1984-2021. Detailed validations of the algorithm can be found in Zhao et al. (2019; 2022) and Tian et al., (2021). The

validation of simulated evaporation at an annual scale from Tian et al. (2022) at 47 reservoirs was summarized in Fig. S11 through a literature review. The results in Fig. S11 indicate that the modeled average annual evaporation rates match well with the observed rates. Specifically, the percent bias (PBIAS), Nash-Sutcliffe efficiency (NSE), and root-mean-square error (RMSE) were found to be 0.02%, 0.82, and 11.2 mm, respectively. This high level of agreement suggests that the Penman method is a reliable approach for calculating reservoir evaporation rates in China. Fig. S12 shows the long-term mean meteorological variables that were used to calculate the evaporation rates.

[Figure]

Figure S11. Observed and modeled average annual evaporation for 47 reservoirs (Tian et al., 2021).

[Figure]

**Figure 8.** Validation of reconstructed monthly reservoir evaporation values. (a) Long-term mean evaporation rates and (b) water surface areas during 1984-2020.


Figure 9 caption: are these just "topographic" characteristics or in general "catchment" characteristics?

R2C11: Yes, we checked it. These are topographic characteristics.

Sections 3.4.2 – 3.4.4: The results and graphics here are nice; however, I wonder what the utility of these data/outcomes are. I suggest that the authors shed some light in the intro section and subsequently in the results section regarding why these specific attributes are chosen, and why/how these are useful, for example, for modeling hydrology considering reservoirs.

R2C12: Our study involved the integration of multiple attributes, offering a good dataset to comprehending the features of reservoir-catchments in China systematically. The Res-CN dataset holds considerable potential in advancing the comprehension of the processes involved in Chinese reservoirs. We have further elaborated on this dataset in the introduction, summary, and applications sections.

**Introduction:** In addition to the time series of reservoir datasets described above, reservoir upstream catchment attributes (e.g., climate, geology & soil, topography, land cover, and anthropogenic activity characteristics) are also important as reservoirs collect materials from upstream catchments. Researchers can better understand catchment-level landscape limnology by incorporating these attributes (Soranno et al., 2010). To promote standardized large sample studies and improve the utility of our Res-CN, we have incorporated catchment attributes initially introduced by Addor et al. (2017) in their Catchment Attributes and MEteorology for Large-sample Studies (CAMELS), as well as numerous follow-up studies such as CAMLES-CL, CMALES-BR, CAMLES-GB, (Alvarez-Garreton et al., 2018; Chagas et al., 2020; Coxon et al., 2020), LamaH-CE (Klingler et al., 2021), CCAM (Hao et al., 2021), LakeALTAS (Lehner et al., 2022), and studies by Chen et al. (2022) and Liu et al. (2022), while additionally including several other attributes. These lake datasets and station-based datasets of catchment characteristics proved that catchment-level attribute datasets are very useful.

[revised manuscript text omitted]

**Section 3.4.5: Again, why are these specific human activities selected for analysis and how are those useful?**

R2C13: Yes, Hope the above R2C12 response addressed your concern.

References:

Liu, J., Fang, P., Que, Y., Zhu, L.-J., Duan, Z., Tang, G., Liu, P., Ji, M., and Liu, Y.: A dataset of lake-catchment characteristics for the Tibetan Plateau, Earth Syst. Sci. Data, 14, 3791–3805, https://doi.org/10.5194/essd-14-3791-2022, 2022.

Related to the above comments on the utility of various characteristics, I would suggest adding one figure on the ratio of reservoir storage and/or surface area to catchment size.

R2C14: Thanks for your reminder! Actually the ratio of reservoir storage and/or surface area to catchment size is already included in the dataset of "topographic characteristics". We have created the figure S13 as suggested.

Main text: Besides, we also added "resArearatio" to describe the proportion of the reservoir water surface area to the catchment area (Fig. S13).

[Figure]

**Figure S13.** Spatial distribution of the ratios of reservoir water surface area and storage to catchment area. Note: not all reservoir water storage data are available from the GeoDAR database (Wang et al., 2022).

Overall/General: the number of reservoirs selected for various purposes is different and validation is provided for a limited subset. Please try to have consistency and expand the validation effort.

R2C15: We apologize for the inadvertent omission of the validation figures for the 138 reservoirs in our Res-CN dataset. We have taken corrective measures by uploading the figures to the same Zenodo link of our Res-CN data product, and we kindly request that you access them from there https://doi.org/10.5281/zenodo.7664489.

Considering the extensive information contained within the supplementary file, we recognize the potential benefits of incorporating the validation figures - which, due to their size, span multiple pages - in our data product to facilitate user access and convenience. However, we also recognize the importance of maintaining a balance between completeness and conciseness in the main text. Consequently, we have presented only a subset of validations for select reservoirs alongside the overall evaluation accuracy. Nevertheless, we would like to assure users that all validation information is available in the data products. We are confident that this balance between completeness and conciseness is in line with the expectations of our readers, and we encourage them to refer to the data products for more detailed information.

The validation figures for all 138 reservoirs can be found in the **"validation_figures"** folder, which includes the time series of reservoir water level, water area, storage variation, and evaporation. In the "water level" directory, the time series of reservoir water level are available in two modes, i.e., high rate product and standard rate, along with their comprehensive evaluation reports and figures in PDF and TXT files. The "water area" directory provides the monthly area time series of reservoirs, accompanied by their comprehensive evaluation Excel files, including CC values compared with satellite-based water level, in situ water level, and other areal time series from other studies. Finally, the "storage variation" directory includes the time series and comprehensive evaluation figures in PDF files, which include statistical metrics.

Thank you for your feedback, and we hope that the inclusion of these validation figures will facilitate the use of our Res-CN dataset.

Minor/Editorial comments:

L48, "...especially driven by climate warming and ...": not clear "what" is driven by climate and population; revisions needed.

We have rephrased as follows:
it is essential to develop a comprehensive publicly available reservoir data set in the context of growing interest of reservoir studies and water managements.

L80: should be "altimetry-based reservoir datasets" and "Chinese reservoirs"

Thanks, we have changed as:

In three popular altimetry-based reservoir datasets (Hydroweb, G-REALM, and DAHITI), there are approximately 30 Chinese reservoirs.

L101: please spell out GEE

Thanks, we have changed as:

GEE (Google Earth Engine)

L108: delete "for"

Thanks, we have deleted it.

Figure 1 caption and elsewhere: I suggest "water SURFACE area" instead of "water area"; this applies to Section 2.2 as well.

Thanks, we have changed as water surface area throughout the paper.

L133: please check grammar.

We have rephrased it:

The Global Surface Water Explorer was used to select altimetric data for which water occurrence is greater than 10% (Zhang et al., 2020).

Figure 3 caption: "dimensionless XX? is indicated ...."

We deleted this sentence, and not show this symbol.

---

## Author Comment (AC3)

**Reviewer #3 Comment on essd-2022-422 (Anonymous Referee #3)**

Dear Anonymous Referee #3,

Thank you for your time and efforts in reviewing our manuscript. Please find attached point-to-point responses regarding your comments (marked in purple) and made corresponding changes in the main manuscript (in red). We hope that the improved manuscript can help the readers to better understand our study.

Kind regards.

**General comments**

Dams and reservoirs play an important role in water resource management and regulation. The authors provided new and comprehensive reservoir datasets over China (the Reservoir dataset in China, Res-CN), which featured reservoir-catchment characteristics for 3254 reservoirs. I have the following concerns for authors in ongoing revision and improvements.

R3C0: Thank you for your recognition of the strengths of our study. We appreciate your constructive feedback and have carefully considered all of your suggestions and comments. We hope that these revisions have improved the overall quality of the manuscript. We are grateful for your time and expertise in reviewing our work, and we believe that your feedback has made a valuable contribution to the study's scientific value. Thank you again for your comments, and we hope that the revised manuscript will meet your expectations.

Authors may need to provide more details on why they only focus on reservoirs in China, and why they chose to use GeoDAR while a more recent study published more comprehensive reservoir dataset for China.

R3C1: Thank for your constructive comments. We have re-organized the introduction, adding more details on the reason why we only focus on Chinese reservoirs, and thanks for altering us about the new data, we have explained it in the Summary, applications and outlook. We acknowledge all great efforts made by our scientific members. We argue the new added reservoir shapefiles are mainly very small reservoirs. Our study aims to provide a comprehensive and extensive dataset of reservoir-catchment characteristics in China for a better understanding of reservoir impacts on hydrological and biochemical cycles, these thousands of very small reservoirs are not included in our study. Thus, Res-CN still shows significant improvements and unique contribution in its comprehensive and complete information. We hope that our datasets will contribute to the development of more effective water quality management strategies for Chinese reservoirs and serve as a

valuable resource for researchers and policymakers in this field. Please find the revised texts below. Hope it addressed your concerns.

[revised manuscript text omitted]

Authors have provided comprehensive climatic characteristics (L450~L451) and human activity characteristics of reservoirs but did not explicitly state why these characteristics should be provided. Therefore, I suggest that the authors to offer a more compelling motivation to start their Introduction, and also discuss why these much informaiton is needed for understanding reservoir changes. Otherwise it may look like too much information to digest for certain users.

R3C2: Our study involved the integration of multiple attributes, offering a good dataset to comprehending the features of reservoir-catchments in China systematically. The Res-CN dataset holds considerable potential in advancing the comprehension of the processes involved in Chinese reservoirs. We have further elaborated on this dataset in the introduction, summary, and applications sections. Hope this addressed your concerns.

[revised manuscript text omitted]

Although authors argued the need for intermediate catchments, however, I still failed to understand how data for these intermediate ones can be used to understand changes in reservoirs as I thought it may be missing essential water balance components? Can authors add more discussions and also clarify?

R3C3: We have added explanations in the Summary, applications, and outlook. Hope the above comments addressed your concern. Taking the first application as an example, we can use the modeled outflow or sediment of upstream reservoirs, and modeled sediment or water mass in the local drainage catchment of the downstream reservoir catchment (i.e., intermediate catchment in Fig 2b) to explore the quantify the relative contributions of upstream reservoirs and local drainage catchment on water quality (e.g., algae contributions and water color) of downstream reservoir by tracking temperature and nutrient flows from upstream reservoirs and intermediate catchments (e.g., Hou et al., 2022; Yang et al., 2022).

We envision that Res-CN with its comprehensive and extensive attributes can provide strong supports to a wide range of applications and disciplines. Firstly, our two types of catchments along with their catchment-level attributes allow investigations within individual catchments and interconnected river networks. For example, as illustrated in Figure 2, users may quantify the relative contributions of upstream reservoirs and local drainage catchment on water quality (e.g., algae contributions and water color) of downstream reservoir by tracking temperature and nutrient flows from upstream reservoirs and intermediate catchments (e.g., Hou et al., 2022; Yang et al., 2022). Besides, water and sediment transfer can be also more accurately simulated in such a spatially explicit context if appropriate approaches are used. Machine-learning methods make it possible to predict reservoir storage change at 1- to 3-month lead from reservoir upstream attributes and time-series of reservoir states (Tiwari et al., 2019).

[Figure]

**a) Basin delineation A | Full catchments**

**b) Basin delineation. B | Intermediate catchments**

**Figure 2.** An example of the types of catchment delineations in Res-CN. (a) Catchment delineation A: full catchments, which are defined as the entire area contributing to a reservoir. In plot (a), full catchment of reservoir 23720 overlaps with that of reservoir 3205 and that of 6651. (b) Catchment delineation B: intermediate catchment. In plot (b), all upstream contributing areas of the upstream reservoirs (3205 and 6651) are removed from the full catchment of reservoir 23720, thus, we get the intermediate catchment of reservoir 23720 (in black boundary). Background in light blue indicates other catchments not shown in this example. Source of background: MERIT Hydro and MERIT DEM (Yamazaki et al., 2019).

**Other Comments**

L243: there's no need to mention computational time, unless you can provide details on the platform because it is highly platform dependent.

Reply: Thank for your comment, we have rephrased it:

This algorithm can correct the river networks by analyzing the gradients of flow accumulations along the rivers and can rapidly delineate catchments.

L365~L370: I suggest that the author should place these introductions after L358 (GRSAD and RealSAT) to make the content more cohesive.

Reply: I completely agree with you and appreciate your constructive feedback. These sentences are positioned after L358 is to enhance the cohesiveness of the overall content.

We compare these datasets with in situ water levels and altimetric measurements as well as other areal datasets (GRSAD and RealSAT). RealSAT generated 681,137 monthly Lake-surface area maps from Landsat images during 1984-2015 using an ORBIT (Ordering-Based Information Transfer) approach that has been validated on 94 large reservoirs. As opposed to RealSAT, which generated new static lake polygons from water occurrence data, GRSAD used existing static surface water polygons, HydroLAKES and GRanD, to create monthly areas for 6,817 global reservoirs based on Landsat images over the last 35 years.

L381~L382: Why the time period of reservoir storage variation is from 1984 to 2020, not 1984~2021?

Reply: Upon careful review of the information, we have identified an error in our manuscript. The correct date range should be 1984-2021 instead of what was previously stated. We apologize for any confusion this may have caused. However, we want to assure you that the rest of the manuscript, including Table 2, contains accurate information. We have corrected the mistake in Line 381-382.

L395: Please explain the two peak values of in-situ in 2021 (Figure.7 and Figure.8).

Reply: For the figure 7: Yes, we add more explanations and discussed the uncertainties as well as limitations in this section. Please also note that for Fig. 7 panels 9-12, our data indeed captured the large peak values for most reservoirs (2, 0.5, 0.2 km$^3$).

Main text: The Res-CN database provides monthly reservoir water storage anomaly for 3254 Chinese reservoirs during 1984-2020 using DEM's area-storage model, along with their detailed evaluation reports (see Section of data availability).

The remotely sensed storage anomalies generally agree with the observations represented by the statistical metrics, although some large discrepancies occur in peak values.

The uncertainties in storage anomalies are primarily attributed to three sources, i.e., the altimetric water level, water surface area estimations from Landsat and Sentinel-2 images, and the error resulting from their combination (the hypsometric curve). Fig. S10 provides an example that illustrates how the uncertainties in satellite datasets propagate to storage anomalies. According to Shen et al. (2022), the primary source of error in storage anomaly is water surface area and the hypsometric curve. Regarding the water surface area, after applying the algorithm developed by Donchyts et al. (2022), these errors and impacts can be reduced to a large

extent. Meanwhile, we employed five hypsometric relationships, and the one with the highest $R^2$ value for further use. For more than 80 % reservoirs, the $R^2$ values are greater than 0.5, providing a strong foundation for storage anomaly estimates. Nonetheless, the current satellite sensors have limitations, as evidenced by the significant discrepancies observed in peak values (Figure 7). The increasing temporal resolution and data accuracy of satellite datasets, such as the SWOT mission, will likely improve the accuracy of storage anomaly estimates in the future.

[Figure]

**Figure 7.** Time series of water surface area and storage anomaly in selected reservoirs. RMSE (km³), NRMSE, and CC values are given at the top of each subplot when in situ observations available. Note that: time series of water surface area and storage anomaly of the remaining reservoirs are available in our datasets.

[Figure]

**Figure S10.** Graphs showing an example that illustrates how the uncertainties in satellite datasets propagate to storage anomalies. Error series and relationships of reservoir elevation-storage. Error series of (a) SWE-derived RWSC (i.e., storage anomaly), (b) WSE-derived RWSC and water level change, (c) WSE (i.e., water level). (d) and (e) Relationships of elevation-storage. The numbers on the x-axis (a, b, c) refer to the IDs of SWE, WSE, and WSE change observations, respectively. For more details about the propagation process, please find the reference Shen et al., (2020): https://doi.org/10.3390/rs14040815.

For Figure 8: We revised the figure 8 and re-create figure s11 for validation. Please find our revised text below:

Res-CN provides monthly reservoir evaporation values for 3254 Chinese reservoirs during 1984-2021. Detailed validations of the algorithm can be found in Zhao et al. (2019; 2022) and Tian et al., (2021). The validation of simulated evaporation at an annual scale from Tian et al. (2022) at 47 reservoirs was summarized in Fig. S11 through a literature review. The results in Fig. S11 indicate that the modeled average annual evaporation rates match well with the observed rates. Specifically, the percent bias (PBIAS), Nash-Sutcliffe efficiency (NSE), and root-mean-square error (RMSE) were found to be 0.02%, 0.82, and 11.2 mm, respectively. This high level of agreement suggests that the Penman method is a reliable approach for calculating reservoir evaporation rates in China. Fig. S12 shows the long-term mean meteorological variables that were used to calculate the evaporation rates.

[Figure]

Figure S11. Observed and modeled average annual evaporation for 47 reservoirs (Tian et al., 2021).

[Figure]

**Figure 8.** Validation of reconstructed monthly reservoir evaporation values. (a) Long-term mean evaporation rates and (b) water surface areas during 1984-2020.

L463~L464: Why the time period is different between here (1990~2018) and L456 (1980~2020)?

Reply: We updated the metrics based on all available data, and revised it:

We calculated nine attributes for NSCD based on meteorological data between 1 October 1990 and 30 September 2020 to reflect aspects of climatic characteristics.

L524: Why the color of Fig.11i is red?

Reply: We have thoroughly examined the figure in question, as well as all the other figures in the manuscript, and have confirmed that it is accurate. We apologize for any confusion or misunderstanding that may have arisen, and we hope that this clarification has helped to address any concerns.

---

## Author Response (AR2)

Dear Handling topical editor Dalei Hao,

Thank you for your prompt and thorough handling of our manuscript "essd-2022-422." We are grateful for your expeditious review and appreciated your swift response to our revision. We have added one more figure in the main txt based on the reviewer 1's comment. Hope the revised manuscript are to your satisfaction.

I found the authors addressed my previous comments well. I only have a minor comment at this time. I appreciate the authors to upload the time series comparisons of water level for all the available reservoirs. But I still think it will be useful to reduce all the plots into one and show it in the main text. For example, the authors can plot the histogram of the evaluation metrics from all the reservoirs similar to Figure 6 in the main text. I understand RMSE is a common evaluation metric, but I still don't think it is a good to compare across different sites (line 455 - line 460). Why not showing the relative RMSE for the water level evaluation? I noticed rBIAS and rRMSE were used in Figure 6 for validating water area, similar evaluations may be done in Figure 4 for validating water level.

Reply: This is indeed a great point and thanks for your comments. We added one more figure in the Supplementary file. Please check it below.

[Figure]

Figure S9. Performance of the Standard-rate (SR) and High-rate (HR) products in terms of the RMSE and the relative RMSE values of the validated reservoirs. For detailed validation metrics, please refer to our data.

Best,

Youjiang Shen.